# A Novel Multi Level Dynamic Decomposition Based Coordinated Control of Electric Vehicles in Multimicrogrids

Muhammad Anique Aslam [1,*], Syed Abdul Rahman Kashif [1], Muhammad Majid Gulzar [2,3], Mohammed Alqahtani [4] and Muhammad Khalid [3,5,6,*]

1 Department of Electrical Engineering, University of Engineering and Technology, Lahore 54890, Pakistan; abdulrahman@uet.edu.pk
2 Department of Control & Instrumentation Engineering, King Fahd University of Petroleum and Minerals, Dhahran 31261, Saudi Arabia; muhammad.gulzar@kfupm.edu.sa
3 Interdisciplinary Research Center for Renewable Energy and Power Systems (IRC-REPS), King Fahd University of Petroleum and Minerals, Dhahran 31261, Saudi Arabia
4 Department of Industrial Engineering, King Khalid University, Abha 62529, Saudi Arabia; m.alqahtani@kku.edu.sa
5 Electrical Engineering Department, King Fahd University of Petroleum and Minerals, Dhahran 31261, Saudi Arabia
6 SDAIA-KFUPM Joint Research Center for Artificial Intelligence, King Fahd University of Petroleum and Minerals, Dhahran 31261, Saudi Arabia
* Correspondence: maniqueaslam@uet.edu.pk (M.A.A.); mkhalid@kfupm.edu.sa (M.K.)

**Abstract:** This paper presents a novel tetra-level dynamic decomposition-based control approach for coordinated operation of electric vehicles in multimicrogrids, which is comprehensive, generic, modular, and secure in nature, to maximize the utilization of renewable energy sources, while meeting the load demands with the resources available. There are a number of microgrids that are connected to the grid. Each microgrid consists of a number of renewable energy sources, energy storage systems, non-renewable energy sources, electric vehicles, and loads. Each distributed energy source or load is controlled by a microsource controller. All microsource controllers with a similar nature are controlled by a unit controller, and all the unit controllers in a microgrid are controlled by a microgrid controller. There is a single multimicrogrid controller at the top. The proposed control scheme was verified through simulation-based case studies.

**Keywords:** microgrid; multimicrogrid; electric vehicle; control of multimicrogrids

## 1. Introduction

Electric vehicles (EVs) are continuously increasing in use as promising alternatives to conventional fossil fuel based vehicles, due to their low pollution and operational cost [1–3]. These advantages can only be gained if the charging of EVs is controlled properly. There is no economical benefit if EVs are not charged smartly. For example, an EV charged from a coal power plant will produce more pollution than a conventional fossil-fuel-based EV [4–8]. On the other hand, EVs with high parking hours (22 on average [9]) act as mobile energy storage that can be deployed to support a conventional grid [10–14]. Thus, it is the need of the hour to devise smart control strategies that can charge EVs, especially from renewable energy sources (RESs) and support the conventional grid. As far as the integration of RESs is concerned, microgrids (MGs) have proven to be the best option [15,16]. A MG consists of distributed energy sources (DESs) and storage that are controlled to supply the assigned load, with or without the conventional grid [17–24]. Proper control of such a system will enhance the penetration of RESs, by removing their intermittency. This can also be utilized for EVs. A number of interconnected MGs form multimicrogrids (MMGs) that provide more flexibility compared to a single MG if controlled properly [25–30].

A number of methods have been proposed in the literature to control the operation of EVs integrated into MGs. Reference [31] presented a double-loop optimized cooperative control scheme for economic dispatch and capacity allocation of RESs and EVs in a MMG environment, using an improved particle swarm optimization algorithm. Reference [32] proposed an optimal control strategy for a MG consisting of solar panels, a wind turbine, pumped hydro storage system, and EVs, using a modified whale-optimization-based metaheuristic algorithm. Reference [33] proposed a modified dragon fly algorithm to minimize the operational cost of a MG consisting of RESs, energy storage systems (ESSs), EVs, and loads. Support vector regression was used to estimate the charging demands of EVs for a period of one day. Reference [34] presented an optimized operation of a Monte Carlo simulation-based MG model consisting of RESs, energy storage units (ESUs), EVs, and loads, considering technical constraints under different uncertainties using a modified sparrow search algorithm. Reference [35] designed an atom-search-optimization-based tilt integral derivative controller, to enhance the load frequency response of a MMG incorporating EVs. In [10], a centralized energy management system was proposed to support faulty MGs by using EVs available in normal MGs, to improve the resiliency of the network. Reference [36] presented a model predictive control of a MG consisting of EVs, microturbines, and a battery-based energy storage system (BESS) in a hierarchical manner. The higher level control was responsible for long-term energy scheduling, while the low level control maintained the load frequency control in the short term. Reference [37] proposed a two-layered control of EVs in MMGs. The top layer is the MMG layer that deals with incoming vehicles, while the bottom layer is the EV layer that controls the charging process. The control is based on a column and constraint generation algorithm for a mixed integer linear program based problem. Uncertainties are dealt with using a neural network data obtained from historical profiles. Reference [38] presented a two-level control of MMGs comprised of EVs using a back propagation neural network forecasting of EV data. In the MMG layer, a multi-objective problem aimed at minimizing transmission losses, operating costs, and carbon emissions is solved with an adaptive multi-objective evolutionary algorithm, while the MG layer uses a consistency algorithm for economic dispatching. In [39], a master slave control was proposed for a power system having integrated RESs, which was capable of operating under faults. The authors in [40] proposed genetic-algorithm-based day ahead scheduling of EVs in MMGs in grid-connected and islanded modes. Reference [41] proposed a two-level control of grid-connected MGs based on particle swarm optimization. First, the grid load is optimized, in order to minimize the total cost and maximize the penetration of RESs and the profit. Second, the EVs are treated as shiftable loads, to satisfy the dispatch of the first layer. Reference [9] presented a two-step control of EVs in MMGs. In the first step, a centralized controller allocates the charging/discharging schedule to the EV aggregator in each MG. In the second step, the EV aggregator schedules each EV under its influence with minimum switching transitions. In [42], the authors presented a three-step approach to using EVs for resilience improvement of MMGs in the event of contingencies. In the first step, a central control acquires the available and required power from EVs, while the power is shared by the grid-connected MGs with their islanded counterparts. In the last step, each MG controls its own EVs to meet the demand of the central controller. Reference [43] also aimed to increase the grid resilience through coordinated operation of EVs in MMGs using day ahead scheduling and a rolling horizon approach. Reference [44] presented a two-level hierarchical control of residential MGs integrated into a distribution network. In [45], a similar control approach was presented for hybrid MGs based on a flower pollination algorithm. Uncertainties were modeled intelligently using historical data. Reference [46] presented a two-level control of MMGs using deep neural networks. The computational burden was reduced by estimating the parameters of MGs, instead of calculating the probabilistic power flow. The authors presented a two-level multi-objective non-linear control for a DC/DC converter connected between a high voltage and low voltage bus in [47]. The low level control tracks the references selected by the high level control. Moreover, ref. [48] presented an adaptive

sliding mode control for achieving multiple control tasks based on supervisor selection. Similarly, ref. [49] used a Lagrange-multiplier-based method for coordinated operation of hydrothermal units in an HVDC environment. Such control strategies can be adapted to integrate EVs into MMGs. Reference [1] proposed a decentralized model predictive control of EVs in bipolar DC MGs using their charging profiles. Similarly, ref. [50] used a fuzzy-logic-based controller to model and control a hybrid power system. Reference [51] proposed a particle swarm and interior point iterative distributed algorithm for optimal operation of MMGs. Similarly, ref. [52] optimized the operating cost of MGs having RESs and ESSs using mixed integer programming with quadratic constraints. Reference [53] proposed a multi-objective-based central control scheme to optimize the power flow in a DC MG consisting of a wind energy source, EV, load, and ESU, to minimize costs and maximize the state of charge (SoC) of the EV. In [54], the estimation of the SoC of EV batteries was presented using Kalman filtering and neural networks. Similarly, ref. [55] used machine learning to estimate the battery degradation of EVs. Reference [56] minimized the cost and voltage fluctuations in a MG using a multi-objective stochastic optimization and ref. [57] presented a similar mixed integer linear programming based control of multi-energy MGs.

It is clear from the above discussion that most researchers have used a centralized control structure among the available control structures (centralized, decentralized, distributed, or hierarchical), with offline centralized control schemes for fixed participants to coordinate the operation of EVs in an MMG. Keeping in view these limitations, this paper presents a novel tetra-level dynamic control approach for coordinated operation of EVs in MMGs. The major contributions of this paper are as follows:

- This paper presents a novel hierarchical control of EVs in MMGs that combines the benefits of both centralized and distributed control [16,58,59];
- The proposed control approach is based on the decomposition principle. This imparts a certain level of independence to the controllers at different levels of the hierarchy. If one controller fails, other controllers continue to work;
- The proposed control scheme is real-time in nature. Thus, it can handle uncertainties;
- The proposed control scheme intends to maximize the utilization of RESs, while meeting the load requirements within the available resources; thus, minimizing the operational costs and maximizing the load satisfaction;
- The proposed control scheme provides a comprehensive control of EVs, keeping MMGs in view with five different modes of operation, as mentioned below:
    - Minimum time charging mode of operation;
    - Minimum time fixed cost charging mode of operation;
    - Minimum cost charging mode of operation;
    - Maximum SoC fixed cost charging mode of operation;
    - Grid support mode of operation;
- The proposed control scheme is generalized with respect to different types of EVs and optimization techniques for their scheduling;
- The proposed control scheme is modular as far as the number of EVs is concerned;
- The proposed control scheme supports plug and play capability for EVs;
- The proposed control scheme makes possible the utilization of simple computational resources with dynamic-decomposition-based hierarchical control;
- The proposed control scheme intends to minimize the exchange of information, to ensure the maximum possible privacy for EVs.

The rest of this paper is laid out as follows: Section 2 presents the proposed control methodology. Section 3 discusses the results of applying the proposed control technique to a MMG. Section 4 concludes the paper, while listing some points for the future work.

## 2. Proposed Control Methodology

Figure 1 shows the structure of the MMG under consideration. There are a number of MGs that are connected to the grid at the point of common coupling (PCC). Each MG consists of a number of RESs, ESSs, EVs, non-renewable energy sources (NRESs), and loads. Each DES or load is controlled by a microsource controller (MC). All MCs with a similar nature are put under a unit controller (UC). Therefore, there are five UCs; namely, a renewable energy source unit controller (RESC), energy storage system unit controller (ESSC), electric vehicle unit controller (EVC), non-renewable energy source unit controller (NRESC), and load unit controller (LC). All UCs are controlled by a MG controller (MGC). There is a single MMG controller (MMGC) at the top.

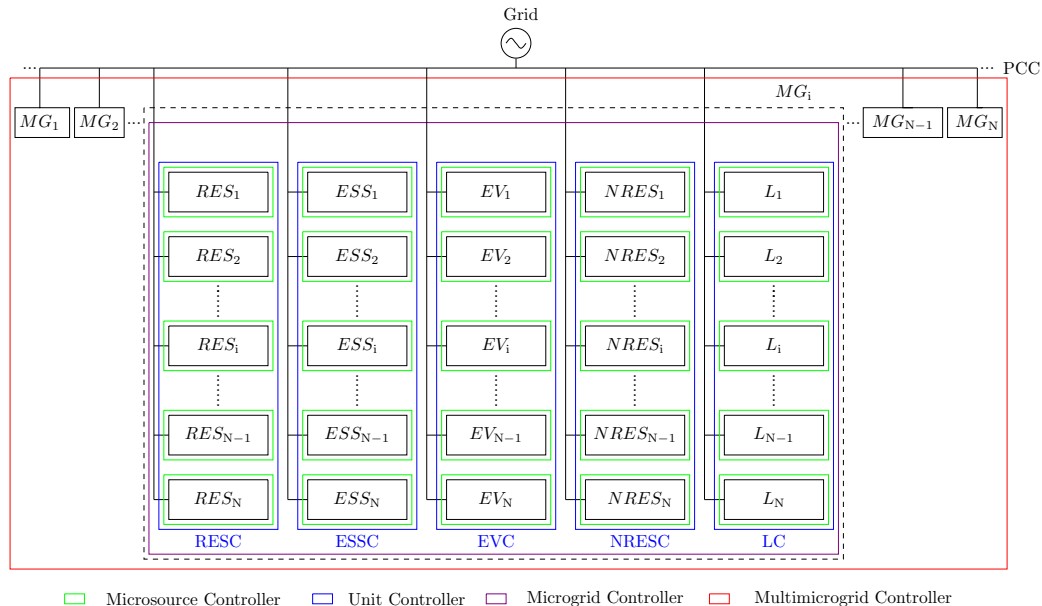

**Figure 1.** Basic control structure of the MMG network. A line diagram of the network is represented in black. The sphere of influence of each hierarchical controller is represented by a rectangular box.

Figure 2 represents the detailed layout of the proposed MMG control structure. Each RES conveys the available apparent power $S_{\text{res,avb,i}}$ to the respective UC. Based on this information, the RESC informs the MGC about the total power available from the RESs ($S_{\text{res,avb,mg,i}}$). Each ESU conveys the upper ($SoC_{\text{ul,ess,i}}$) and lower ($SoC_{\text{ll,ess,i}}$) limits of SoC and rated power ($S_{\text{ess,rtd,i}}$) to its UC, which calculates the current SoC ($SoC_{\text{ess,i}}$) and power of the units capable of charging ($S_{\text{ess,co,avb,mg,i}}$), discharging ($S_{\text{ess,do,avb,mg,i}}$), or both ($S_{\text{ess,cd,avb,mg,i}}$) in the respective MG. Each EV informs about its capacity ($C_{\text{ev,i}}$), rated power ($P_{\text{r,ev,i}}$), initial ($SoC_{\text{i,ev,i}}$), final ($SoC_{\text{f,ev,i}}$), minimum ($SoC_{\text{min,ev,i}}$) and maximum ($SoC_{\text{max,ev,i}}$) SoC, payable cost ($cost_{\text{ev,i}}$), and mode of operation ($m_{\text{ev,i}}$), along with the arrival ($t_{\text{a,ev,i}}$) and departure ($t_{\text{d,ev,i}}$) times. EVC performs dynamic scheduling of EVs accordingly and conveys the required or available power of each EV ($P_{\text{ev,i}}$) to the EVC. The NRESC takes the available rated power of each NRES ($S_{\text{nres,rtd,i}}$) to its MGC. The LC informs its MGC about the total load of the MG. The MGC, having all the available and required powers, decides whether to offer and/or request from the other MGs via the MMGC. The MMGC receives information about the available and required resources from each MG and schedules the most suitable resources of a MG to supply the other MG/s, where needed. Such a MMG control system is versatile in operation, generalized as far as the MGs and microsources are concerned, modular as far as the number of participants is concerned, and secure as far as the exchange of information is concerned. Versatility or operational comprehensiveness results from the fact that each DES can be controlled in its own particular way or ways by means of a separate MC, which then aggregate with similar MCs in the form of an UC. Thus, there are no restrictions from other participants

on the way a particular MC operates. This fact will become more evident when we present the different modes of operation of an EV in Section 2.3. The generalization characteristic results from the fact that the control parameters do not depend on the structure of the participating entities. The modularity stems from the fact that the number of EVs can increase or decrease. The security results from the fact that particular information is shared with the respective unit controller only. Above the unit controller level, only the available and required power information is conveyed or received at an aggregated level. This makes the local DES unit information unavailable to the controllers at MG level. Similarly no information about the type, structure, or nature of the participating MG is conveyed to the MMGC.

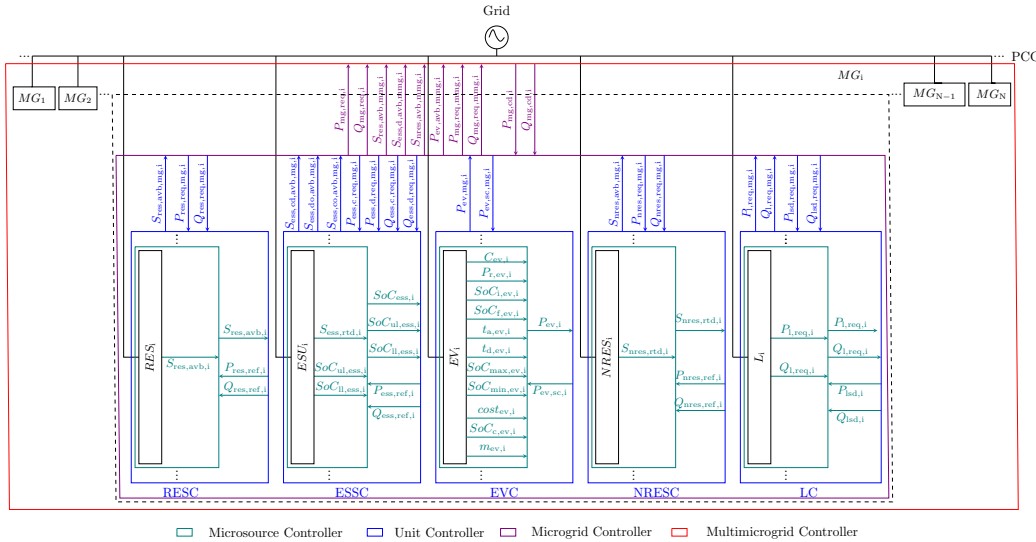

**Figure 2.** Detailed layout of the control scheme.

In the following, we describe the control algorithms for the MMGC, MGC, and EVC, which collectively coordinate in a dynamic and hierarchically decomposable manner, to control the operation of the EVs in a MMG.

## 2.1. Multimicrogrid Controller (MMGC)

The MMGC receives information about the available power from each resource type (i.e., RESs, NRESs, ESUs, and EVs) that the particular MGC makes available for exchange with other MGs, as well as the load (active and reactive) that the particular MGC is not able to supply from its own resources. The MMGC checks each MG for one of the following three cases (see Algorithm 1):

In the case of an active power requirement alone, the MMGC allocates power from the available resources in a predefined order, in such a way that the resources from the nearby MG are used preferably. This can be achieved by following the steps mentioned below:

---

**Algorithm 1** Proposed MMGC Algorithm

---

Construct the MMG matrix ($M_{\mathrm{mmg}}$) with $i^{th}$ row having $S_{\mathrm{res,avb,mmg},i}$, $P_{\mathrm{ev,avb,mmg},i}$, $S_{\mathrm{ess,d,avb,mmg},i}$, $S_{\mathrm{nres,avb,mmg},i}$, $P_{\mathrm{req,mg},i}$ and $Q_{\mathrm{req,mg},i}$
**for** for each microgrid $i$ **do**
    **if** only active power is required **then**
        **if** $S_{\mathrm{res,avb,mmg}} \geq M_{\mathrm{mmg}}(i,5)$ **then**
            Check the nearby MGs one by one
            **if** $M_{\mathrm{mmg}}(N_{\mathrm{nmg}},1) \geq M_{\mathrm{mmg}}(i,5)$ **then**
                Operate the required portion of RESs of the nearby MG available at MMG level to supply the MG under consideration
            **else**
                Operate all RESs of the nearby MG available at MMG level to supply the MG under consideration
            **end if**
        **else if** $S_{\mathrm{res,avb,mmg}} + P_{\mathrm{ev,avb,mmg}} \geq M_{\mathrm{mmg}}(i,5)$ **then**
            Operate all RESs of all the nearby MGs available at MMG level to supply the MG under consideration
            Check the nearby MGs one by one
            **if** $M_{\mathrm{mmg}}(N_{\mathrm{nmg}},2) \geq M_{\mathrm{mmg}}(i,5)$ **then**
                Operate the required portion of EVs of the nearby MG available at MMG level to supply the MG under consideration
             **else**
                Operate all EVs of the nearby MG available at MMG level to supply the MG under consideration
            **end if**
        **else if** $S_{\mathrm{res,avb,mmg}} + P_{\mathrm{ev,avb,mmg}} + S_{\mathrm{ess,avb,mmg}} \geq M_{\mathrm{mmg}}(i,5)$ **then**
            Operate all RESs and EVs of all the nearby MGs available at MMG level to supply the MG under consideration
            Check the nearby MGs one by one
            **if** $M_{\mathrm{mmg}}(N_{\mathrm{nmg}},3) \geq M_{\mathrm{mmg}}(i,5)$ **then**
                Operate the required portion of ESUs of the nearby MG available at MMG level to supply the MG under consideration
             **else**
                Operate all ESUs of the nearby MG available at MMG level to supply the MG under consideration
            **end if**
        **else if** $S_{\mathrm{res,avb,mmg}} + P_{\mathrm{ev,avb,mmg}} + S_{\mathrm{ess,avb,mmg}} + S_{\mathrm{nres,avb,mmg}} \geq N_{\mathrm{nmg}}(i,5)$ **then**
            Operate all RESs, EVs and ESUs of all the nearby MGs available at MMG level to supply the MG under consideration
            Check the nearby MGs one by one
            **if** $M_{\mathrm{mmg}}(N_{\mathrm{nmg}},4) \geq M_{\mathrm{mmg}}(i,5)$ **then**
                Operate the required portion of NRESs of the nearby MG available at MMG level to supply the MG under consideration
             **else**
                Operate all NRESs of the nearby MG available at MMG level to supply the MG under consideration
            **end if**
        **else**
            Operate all RESs, EVs, ESUs and NRESs of all the nearby MGs available at MMG level to supply the MG under consideration.
        **end if**
    **else if** only reactive power is required **then**
        **if** $S_{\mathrm{res,avb,mmg}} \geq M_{\mathrm{mmg}}(i,6)$ **then**
            Check the nearby MGs one by one
            **if** $M_{\mathrm{mmg}}(N_{\mathrm{nmg}},1) \geq M_{\mathrm{mmg}}(i,6)$ **then**
                Operate the required portion of RESs of the nearby MG available at MMG level to supply the MG under consideration
             **else**
                Operate all RESs of the nearby MG available at MMG level to supply the MG under consideration
            **end if**
        **else if** $S_{\mathrm{res,avb,mmg}} + S_{\mathrm{ess,avb,mmg}} \geq M_{\mathrm{mmg}}(i,6)$ **then**
            Operate all RESs of all the nearby MGs available at MMG level to supply the MG under consideration
            Check the nearby MGs one by one
            **if** $M_{\mathrm{mmg}}(N_{\mathrm{nmg}},3) \geq M_{\mathrm{mmg}}(i,6)$ **then**
                Operate the required portion of ESUs of the nearby MG available at MMG level to supply the MG under consideration
             **else**
                Operate all ESUs of the nearby MG available at MMG level to supply the MG under consideration
            **end if**
        **else if** $S_{\mathrm{res,avb,mmg}} + S_{\mathrm{ess,avb,mmg}} + S_{\mathrm{nres,avb,mmg}} \geq N_{\mathrm{nmg}}(i,6)$ **then**
            Operate all RESs and ESUs of all the nearby MGs available at MMG level to supply the MG under consideration
            Check the nearby MGs one by one
            **if** $M_{\mathrm{mmg}}(N_{\mathrm{nmg}},4) \geq M_{\mathrm{mmg}}(i,6)$ **then**
                Operate the required portion of NRESs of the nearby MG available at MMG level to supply the MG under consideration
             **else**
                Operate all NRESs of the nearby MG available at MMG level to supply the MG under consideration
            **end if**
        **else**
            Operate all RESs, ESUs and NRESs of all the nearby MGs available at MMG level to supply the MG under consideration.
        **end if**
    **else if** both active and reactive powers are required **then**
        **if** $S_{\mathrm{res,avb,mmg}} \geq \sqrt{((M_{\mathrm{mmg}}(i,5))^2 + (M_{\mathrm{mmg}}(i,6))^2)}$ **then**

Check the narby MGs one by one
**if** $M_{\mathrm{mmg}}(N_{\mathrm{nmg}},1) \geq \sqrt{M_{\mathrm{mmg}}(i,5)^2 + (M_{\mathrm{mmg}}(i,6))^2}$ **then**
    Operate the required portion of RESs of the nearby MG available at MMG level to supply the MG under consideration
**else**
    Operate RESs of the nearby MG available at MMG level at the load power factor
**end if**
**else**
Operate RESs of all the nearby MGs available at MMG level at the load power factor
**if** $P_{\mathrm{ev,avb,mmg}} \geq M_{\mathrm{mmg}}(i,5)$ **then**
    Check the nearby MGs one by one
    **if** $M_{\mathrm{mmg}}(N_{\mathrm{nmg}},5) \geq M_{\mathrm{mmg}}(i,5)$ **then**
        Operate the required portion of EV resources of the nearby MG available at MMG level to supply the MG under consideration
    **else**
        Operate EVs of the nearby MG available at MMG level to supply the MG under consideration
    **end if**
    **if** $S_{\mathrm{ess,avb,mmg}} \geq M_{\mathrm{mmg}}(i,6)$ **then**
        Check the nearby MGs one by one
        **if** $M_{\mathrm{mmg}}(N_{\mathrm{nmg}},3) \geq M_{\mathrm{mmg}}(i,6)$ **then**
            Operate the required portion of ESS of the nearby MG available at MMG level to supply the reactive power to the MG under consideration
        **else**
            Operate ESSs of the nearby MG available at MMG level to supply the reactive power to the MG under consideration
        **end if**
    **else if** $S_{\mathrm{ess,avb,mmg}} + S_{\mathrm{nres,avb,mmg}} \geq M_{\mathrm{mmg}}(i,6)$ **then**
        Operate ESSs of all the nearby MGs available at MMG level to supply the reactive power to the MG under consideration
        Check the nearby MGs one by one
        **if** $M_{\mathrm{mmg}}(N_{\mathrm{nmg}},4) \geq M_{\mathrm{mmg}}(i,6)$ **then**
            Operate the required portion of NREs of the nearby MG available at MMG level to supply the reactive power to the MG under consideration
        **else**
            Operate all the NRESs of the nearby MG available at MMG level to supply the reactive power to the MG under consideration
        **end if**
    **else**
        Operate all the ESSs and NRESs of the nearby MG available at MMG level to supply the reactive power to the MG under consideration
    **end if**
**else**
Operate all the EVs of all the nearby MGs available at MMG level to supply the active power to the MG under consideration
**if** $S_{\mathrm{ess,avb,mmg}} \geq \sqrt{(M_{\mathrm{mmg}}(i,5))^2 + (M_{\mathrm{mmg}}(i,6))^2}$ **then**
    Check the nearby MGs one by one
    **if** $M_{\mathrm{mmg}}(N_{\mathrm{nmg}},3) > 0$ **then**
        **if** $M_{\mathrm{mmg}}(N_{\mathrm{nmg}},3) \geq \sqrt{M_{\mathrm{mmg}}(i,5)^2 + (M_{\mathrm{mmg}}(i,6))^2}$ **then**
            Operate the required portion of ESUs of the nearby MG available at MMG level to supply the MG under consideration
        **else**
            Operate ESUs of the nearby MG available at MMG level at the load power factor
        **end if**
    **end if**
**else**
Operate all the ESSs of all the nearby MGs available at MMG level to supply the the MG under consideration
**if** $S_{\mathrm{nres,avb,mmg}} \geq \sqrt{(M_{\mathrm{mmg}}(i,5))^2 + (M_{\mathrm{mmg}}(i,6))^2}$ **then**
    Check nearby MGs one by one
    **if** $M_{\mathrm{mmg}}(N_{\mathrm{nmg}},4) \geq \sqrt{(M_{\mathrm{mmg}}(i,5))^2 + (M_{\mathrm{mmg}}(i,6))^2}$ **then**
        Operate the required portion of NRESs of the nearby MG available at MMG level to supply the MG under consideration
    **else**
        Operate NRESs of the nearby MG available at MMG level at the load power factor
    **end if**
**else**
Operate NRESs of all the nearby MGs available at MMG level at the load power factor
    **end if**
    **end if**
    **end if**
    **end if**
**end for**

- If the RESs available at the MMGC are sufficient to provide the deficiency of the MG under consideration, the MMGC allocates RESs of nearby MGs to satisfy the demand. First of all, the closest MG is considered. If it is able to meet the demand, its coordination power is increased by a value equal to the required power (i.e., $P_{\mathrm{mg,cd,N_{nmg}}} = P_{\mathrm{mg,cd,N_{nmg}}} + M_{\mathrm{mmg}}(i, 5)$) and the coordination power of the MG under consideration is decreased by the same amount (i.e., $P_{\mathrm{mg,cd,i}} = P_{\mathrm{mg,cd,i}} - M_{\mathrm{mmg}}(i, 5)$). Here, $N_{\mathrm{nmg}}$ represents $N^{th}$ the nearby MG. In this way, the MGC of the MG under consideration is relieved from dispatching the unsatisfiable load and that load is shifted towards the nearby MG that has sufficient RESs to supply the required load. If the nearby MG is unable to completely supply the required load, its coordination power is increased by an amount equal to the available RESs (i.e., $P_{\mathrm{mg,cd,N_{nmg}}} = P_{\mathrm{mg,cd,N_{nmg}}} + M_{\mathrm{mmg}}(N_{\mathrm{nmg}}, 1)$) and the coordination power of the MG under consideration is decreased by the same amount (i.e., $P_{\mathrm{mg,cd,i}} = P_{\mathrm{mg,cd,i}} - M_{\mathrm{mmg}}(N_{\mathrm{nmg}}, 1)$). After that, the MMGC goes to the next available MG, to see if it has the ability to satisfy the remaining load demand. This process goes on until the required load is satisfied.
- If the required load cannot be satisfied by the RESs available at the MMGC level, but RESs and EVs combined are able to provide the required load (i.e., $S_{\mathrm{res,avb,mmg}} + P_{\mathrm{ev,avb,mmg}} \geq M_{\mathrm{mmg}}(i, 5)$), the MMGC orders the MGCs of all the MGs to dispatch RESs that can be made available at MMG level. This is achieved by decreasing $P_{\mathrm{mg,cd,i}}$ by an amount equal to the $S_{\mathrm{res,avb,mmg}}$ i.e., $P_{\mathrm{mg,cd,i}} = P_{\mathrm{mg,cd,i}} - S_{\mathrm{res,avb,mmg}}$. The coordination powers of all the other MGs are increased by their respective RES capacities available at the MMG. After that, the EVs are dispatched in order of their proximity, to satisfy the remaining load. The closest MG is considered first to supply the load and, if capable, its coordination power is increased accordingly (i.e., $P_{\mathrm{mg,cd,N_{nmg}}} = P_{\mathrm{mg,cd,N_{nmg}}} + M_{\mathrm{mmg}}(i, 5)$) and the coordination power of the MG under consideration is decreased accordingly (i.e., $P_{\mathrm{mg,cd,i}} = P_{\mathrm{mg,cd,i}} - M_{\mathrm{mmg}}(i, 5)$). If the EVs in the nearby MG are unable to supply the required load completely, all the EVs in that MG are dispatched to their full capacity and the next closest MG is considered. This is achieved by increasing the coordination power of the closest MG by $M_{\mathrm{mmg}}(N_{\mathrm{nmg}}, 2)$ (i.e., $P_{\mathrm{mg,cd,N_{nmg}}} = P_{\mathrm{mg,cd,N_{nmg}}} + M_{\mathrm{mmg}}(N_{\mathrm{nmg}}, 2)$) and decreasing the coordination power of the MG under consideration by $M_{\mathrm{mmg}}(N_{\mathrm{nmg}}, 2)$ (i.e., $P_{\mathrm{mg,cd,i}} = P_{\mathrm{mg,cd,i}} - M_{\mathrm{mmg}}(N_{\mathrm{nmg}}, 2)$). This process goes on and until the MG under consideration is supplied to its requirements.
- If the required load cannot be satisfied by RESs and EVs, but the incorporation of the ESSs along with RESs and EVs is sufficient to supply the load (i.e., $S_{\mathrm{res,avb,mmg}} + P_{\mathrm{ev,avb,mmg}} + S_{\mathrm{ess,avb,mmg}} \geq M_{\mathrm{mmg}}(i, 5)$), the MMGC coordinates all the MGCs to fully dispatch their RESs and EVs. The coordination power of the MG under consideration is decreased accordingly (i.e., $P_{\mathrm{mg,cd,i}} = P_{\mathrm{mg,cd,i}} - S_{\mathrm{res,avb,mmg}} - P_{\mathrm{ev,avb,mmg}}$) and the coordination power of each of the nearby MGs is increased by an amount equal to the RES and EV resources available at MMG level (i.e., $P_{\mathrm{mg,cd,N_{nmg}}} = P_{\mathrm{mg,cd,N_{nmg}}} + M_{\mathrm{mmg}}(N_{\mathrm{nmg}}, 1) + M_{\mathrm{mmg}}(N_{\mathrm{nmg}}, 2)$). After that, the MMGC checks the available ESS capacity from each MGC, in order of their geographical proximity. If the nearby MG is able to supply the difference, its MGC is dispatched accordingly. This is achieved by decreasing the coordination power of the MG under consideration by an amount equal to its requirement (i.e., $P_{\mathrm{mg,cd,i}} = P_{\mathrm{mg,cd,i}} - M_{\mathrm{mmg}}(i, 5)$) and increasing the coordination power of the nearby MG by the same amount (i.e., $P_{\mathrm{mg,cd,N_{nmg}}} = P_{\mathrm{mg,cd,N_{nmg}}} + M_{\mathrm{mmg}}(i, 5)$). Otherwise, the MMGC allocates the available portion of the ESUs of the nearby MG to the MG under consideration. This is achieved by decreasing the coordination power of the MG under consideration by an amount equal to the ESU resources of the nearby MG available for MMG interaction (i.e., $P_{\mathrm{mg,cd,i}} = P_{\mathrm{mg,cd,i}} - M_{\mathrm{mmg}}(N_{\mathrm{nmg}}, 3)$) and increasing the coordination power of the nearby MG by the same amount (i.e., $P_{\mathrm{mg,cd,N_{nmg}}} = P_{\mathrm{mg,cd,N_{nmg}}} + M_{\mathrm{mmg}}(N_{\mathrm{nmg}}, 3)$). This process goes on until the load is satisfied.

- If the RESs, EVs, and ESSs available at MMG level are not able to supply the load, but the inclusion of NRESs available at MMG level is sufficient to satisfy the required load (i.e., $S_{\mathrm{res,avb,mmg}} + P_{\mathrm{ev,avb,mmg}} + S_{\mathrm{ess,avb,mmg}} + S_{\mathrm{nres,avb,mmg}} \geq N_{\mathrm{nmg}}(i,5)$), the MMGC asks the MGC of all MGs to completely dispatch the MMG available capacity of their RESs, EVs, and ESSs. The coordination power of the MG under consideration is decreased accordingly (i.e., $P_{\mathrm{mg,cd,i}} = P_{\mathrm{mg,cd,i}} - S_{\mathrm{res,avb,mmg}} - P_{\mathrm{ev,avb,mmg}} - S_{\mathrm{ess,avb,mmg}}$) and the coordination power of each of the nearby MGs is increased by an amount equal to the RES, EV, and ESU resources available at MMG level (i.e., $P_{\mathrm{mg,cd,N_{nmg}}} = P_{\mathrm{mg,cd,N_{nmg}}} + M_{\mathrm{mmg}}(N_{\mathrm{nmg}},1) + M_{\mathrm{mmg}}(N_{\mathrm{nmg}},2) + M_{\mathrm{mmg}}(N_{\mathrm{nmg}},3)$). After that, the MMGC checks the available NRES capacity from each MGC in order of their geographical proximity. If a nearby MG is able to supply the difference, its MGC is dispatched accordingly. This is achieved by decreasing the coordination power of the MG under consideration by an amount equal to its requirements (i.e., $P_{\mathrm{mg,cd,i}} = P_{\mathrm{mg,cd,i}} - M_{\mathrm{mmg}}(i,5)$) and increasing the coordination power of the nearby MG by the same amount (i.e., $P_{\mathrm{mg,cd,N_{nmg}}} = P_{\mathrm{mg,cd,N_{nmg}}} + M_{\mathrm{mmg}}(i,5)$). Otherwise, the MMGC allocates the available portion of the NRESs of the nearby MGs to the MG under consideration. This is achieved by decreasing the coordination power of the MG under consideration by an amount equal to the NRES resources of the nearby MG available for MMG interaction (i.e., $P_{\mathrm{mg,cd,i}} = P_{\mathrm{mg,cd,i}} - M_{\mathrm{mmg}}(N_{\mathrm{nmg}},4)$) and increasing the coordination power of the nearby MG by the same amount (i.e., $P_{\mathrm{mg,cd,N_{nmg}}} = P_{\mathrm{mg,cd,N_{nmg}}} + M_{\mathrm{mmg}}(N_{\mathrm{nmg}},4)$). This process goes on until the load is satisfied.
- If all the resources available at MMG level are not sufficient to supply the load, the MMGC orders the MGC to dispatch the available capacity of RESs, EVs, ESSs, and NRESs and the rest of the load is shed by the respective MGC. This is achieved by decreasing the coordination power of the MG under consideration by an amount equal to the capacity of the RESs, EVs, ESUs, and NRESs available at MMG level (i.e., $P_{\mathrm{mg,cd,i}} = P_{\mathrm{mg,cd,i}} - S_{\mathrm{res,avb,mmg}} - P_{\mathrm{ev,avb,mmg}} - S_{\mathrm{ess,avb,mmg}} - S_{\mathrm{nres,avb,mmg}}$) and increasing the coordination power of the nearby MG by an amount equal to the capacity of the RESs, EVs, ESUs, and NRESs available at MMG level (i.e., $P_{\mathrm{mg,cd,N_{nmg}}} = P_{\mathrm{mg,cd,N_{nmg}}} + M_{\mathrm{mmg}}(N_{\mathrm{nmg}},1) + M_{\mathrm{mmg}}(N_{\mathrm{nmg}},2) + M_{\mathrm{mmg}}(N_{\mathrm{nmg}},3) + M_{\mathrm{mmg}}(N_{\mathrm{nmg}},4)$).

In the case of a reactive power requirement alone, the MMGC allocates power from the available resources in a predefined order, in such a way that the resources from the closest MG are used preferably. This can be achieved by following the steps mentioned above, with the exception that EVs are not available for reactive power support.

The following scenarios arise when both active and reactive power are required.

- If the RESs available at MMG level are sufficient to provide the required power (i.e., $S_{\mathrm{res,avb,mmg}} \geq \sqrt{(M_{\mathrm{mmg}}(i,5))^2 + (M_{\mathrm{mmg}}(i,6))^2}$), the MMGC checks the nearby MGs one by one. If the RESs available for the MMG level of a particular nearby MG are sufficient to meet the load demands, the MGC of that MG is instructed to increase the load by an amount equal to the load required. This is achieved by increasing the coordination powers of the nearby MG to a value equal to the load required (i.e., $P_{\mathrm{mg,cd,N_{nmg}}} = P_{\mathrm{mg,cd,N_{nmg}}} + M_{\mathrm{mmg}}(i,5)$, $Q_{\mathrm{mg,cd,N_{nmg}}} = Q_{\mathrm{mg,cd,N_{nmg}}} + M_{\mathrm{mmg}}(i,6)$) and decreasing the coordination powers of the MG under consideration by the same value (i.e., $P_{\mathrm{mg,cd,i}} = P_{\mathrm{mg,cd,i}} - M_{\mathrm{mmg}}(i,5)$, $Q_{\mathrm{mg,cd,i}} = Q_{\mathrm{mg,cd,i}} - M_{\mathrm{mmg}}(i,6)$). If this is not the case, the coordination powers of the nearby MG are increased by a ratio equal to the power factor of the required load (see Equations (1) and (2)).

$$P_{\mathrm{mg,cd,N_{nmg}}} = P_{\mathrm{mg,cd,N_{nmg}}} + M_{\mathrm{mmg}}(N_{\mathrm{nmg}},1) \times \sqrt{1 + (M_{\mathrm{mmg}}(i,6)/M_{\mathrm{mmg}}(i,5))^2} \quad (1)$$

$$Q_{\mathrm{mg,cd,N_{nmg}}} = Q_{\mathrm{mg,cd,N_{nmg}}} + M_{\mathrm{mmg}}(N_{\mathrm{nmg}},1) \times \sqrt{1 + (M_{\mathrm{mmg}}(i,5)/M_{\mathrm{mmg}}(i,6))^2} \quad (2)$$

The coordination powers of the MG under consideration are decreased by the same value (see Equations (3) and (4)).

$$P_{\text{mg,cd,i}} = P_{\text{mg,cd,i}} - M_{\text{mmg}}(N_{\text{nmg}}, 1) \times \sqrt{1 + (M_{\text{mmg}}(i,6)/M_{\text{mmg}}(i,5))^2} \quad (3)$$

$$Q_{\text{mg,cd,i}} = Q_{\text{mg,cd,i}} - M_{\text{mmg}}(N_{\text{nmg}}, 1) \times \sqrt{1 + (M_{\text{mmg}}(i,5)/M_{\text{mmg}}(i,6))^2} \quad (4)$$

In this way, all the contributing MGs supply the active and reactive powers with a uniform ratio.

- If the RESs are not sufficient to supply the required power, all the RESs available from the respective MGC are dispatched at the load power factor. This is achieved by setting the coordination powers of the nearby MGs according to Equations (5) and (6), and the coordination powers of the MG under consideration according to Equations (7) and (8).

$$P_{\text{mg,cd,N}_{\text{nmg}}} = P_{\text{mg,cd,N}_{\text{nmg}}} - M_{\text{mmg}}(N_{\text{nmg}}, 1) \times \sqrt{1 + (M_{\text{mmg}}(i,6)/M_{\text{mmg}}(i,5))^2} \quad (5)$$

$$Q_{\text{mg,cd,N}_{\text{nmg}}} = Q_{\text{mg,cd,N}_{\text{nmg}}} - M_{\text{mmg}}(N_{\text{nmg}}, 1) \times \sqrt{1 + (M_{\text{mmg}}(i,5)/M_{\text{mmg}}(i,6))^2} \quad (6)$$

$$P_{\text{mg,cd,i}} = P_{\text{mg,cd,i}} - S_{\text{res,avb,mmg}} \times \sqrt{1 + (M_{\text{mmg}}(i,6)/M_{\text{mmg}}(i,5))^2} \quad (7)$$

$$Q_{\text{mg,cd,i}} = Q_{\text{mg,cd,i}} - S_{\text{res,avb,mmg}} \times \sqrt{1 + (M_{\text{mmg}}(i,5)/M_{\text{mmg}}(i,6))^2} \quad (8)$$

After using the RESs, the MMGC moves to EVs. If the EVs available for the MMG level of a particular nearby MG are sufficient to meet the active load demands, the coordination power of the nearby MG is increased to a value equal to the load required (i.e., $P_{\text{mg,cd,N}_{\text{nmg}}} = P_{\text{mg,cd,N}_{\text{nmg}}} + M_{\text{mmg}}(i,5)$ ) and the coordination power of the MG under consideration is decreased by the same value (i.e., $P_{\text{mg,cd,i}} = P_{\text{mg,cd,i}} - M_{\text{mmg}}(i,5)$). If this is not the case, the entire capacity of EVs available at MMG level is deployed to meet the active load demand of the MG under consideration. This is achieved by increasing the coordination power of the nearby MG by a value equal to the available EV resources (i.e., $P_{\text{mg,cd,N}_{\text{nmg}}} = P_{\text{mg,cd,N}_{\text{nmg}}} + M_{\text{mmg}}(N_{\text{nmg}},2)$) and decreasing the coordination power of the MG under consideration by the same value (i.e., $P_{\text{mg,cd,i}} = P_{\text{mg,cd,i}} - M_{\text{mmg}}(N_{\text{nmg}},2)$). In this way, the active load requirements are fulfilled and the MMGC moves towards reactive power scheduling. Since the RESs are depleted, the MMGC checks the ESSs and NRESs for reactive power support. If the ESSs are sufficient to meet the requirements, the ESSs of the nearby MGs are dispatched preferably from the nearby MGs. The coordination power of the nearby MG is increased to a value equal to the load required (i.e., $Q_{\text{mg,cd,N}_{\text{nmg}}} = Q_{\text{mg,cd,N}_{\text{nmg}}} + M_{\text{mmg}}(i,6)$), and the coordination power of the MG under consideration is decreased by the same value (i.e., $Q_{\text{mg,cd,i}} = Q_{\text{mg,cd,i}} - M_{\text{mmg}}(i,6)$). If the ESSs are not sufficient, all the ESUs are dispatched by increasing the coordination power of the nearby MG by a value equal to the available ESS resources (i.e., $Q_{\text{mg,cd,N}_{\text{nmg}}} = Q_{\text{mg,cd,N}_{\text{nmg}}} + M_{\text{mmg}}(N_{\text{nmg}},3)$) and decreasing the coordination power of the MG under consideration by the same value (i.e., $Q_{\text{mg,cd,i}} = Q_{\text{mg,cd,i}} - M_{\text{mmg}}(N_{\text{nmg}},3)$). If the ESSs are not sufficient to meet the load requirements alone, the MMGC checks the NRESs available from the MGCs. If the ESSs and NRESs are both able to meet the requirements, the ESUs of all the nearby MGs are fully utilized to supply reactive power to the MG under consideration. This is achieved by increasing the coordination power of each MG by a value equal to the ESS resources available at MMG level (i.e., $Q_{\text{mg,cd,N}_{\text{nmg}}} = Q_{\text{mg,cd,N}_{\text{nmg}}} + M_{\text{mmg}}(i,3)$) and decreasing the coordination power of the MG under consideration by the ESS resources available

at the MMG level (i.e., $Q_{\mathrm{mg,cd,i}} = Q_{\mathrm{mg,cd,i}} - S_{\mathrm{ess,avb,mmg}}$). After that, the MGCs of the nearby MGs are asked to dispatch the required NRES capacity. If the nearby MG has enough NRESs to meet the requirement, its coordination power is increased by a value equal to the load required (i.e., $Q_{\mathrm{mg,cd,N_{nmg}}} = Q_{\mathrm{mg,cd,N_{nmg}}} + M_{\mathrm{mmg}}(i,6)$) and the coordination power of the MG under consideration is decreased by the same value (i.e., $Q_{\mathrm{mg,cd,i}} = Q_{\mathrm{mg,cd,i}} - M_{\mathrm{mmg}}(i,6)$ ). Otherwise, all NRESs available at MMG level are dispatched by increasing the coordination powers of the nearby MG by a value equal to the available NRESs resources (i.e., $Q_{\mathrm{mg,cd,N_{nmg}}} = Q_{\mathrm{mg,cd,N_{nmg}}} + M_{\mathrm{mmg}}(N_{\mathrm{nmg}},4)$ ) and decreasing the coordination power of the MG under consideration by the same value (i.e., $Q_{\mathrm{mg,cd,i}} = Q_{\mathrm{mg,cd,i}} - M_{\mathrm{mmg}}(N_{\mathrm{nmg}},4)$). If both the ESSs and NRESs are insufficient to meet the demand, all the ESSs and NRESs from all the MGs are dispatched by increasing the coordination powers of the nearby MG by a value equal to the available ESS and NRES resources (i.e., $Q_{\mathrm{mg,cd,N_{nmg}}} = Q_{\mathrm{mg,cd,N_{nmg}}} + M_{\mathrm{mmg}}(N_{\mathrm{nmg}},3) + M_{\mathrm{mmg}}(N_{\mathrm{nmg}},4)$ ) and decreasing the coordination power of the MG under consideration by the same value (i.e., $Q_{\mathrm{mg,cd,i}} = Q_{\mathrm{mg,cd,i}} - M_{\mathrm{mmg}}(N_{\mathrm{nmg}},3) - M_{\mathrm{mmg}}(N_{\mathrm{nmg}},4)$) and the MG under consideration has to shed the remaining load.

- If both the RESs and EVs are unable to meet the requirements, the MMGC dispatches all the available resources of the RESs and EVs from the respective MGCs and checks the ESUs available at MMG level. If these are sufficient to meet the requirements (i.e., $S_{\mathrm{ess,avb,mmg}} \geq \sqrt{(M_{\mathrm{mmg}}(i,5))^2 + (M_{\mathrm{mmg}}(i,6))^2}$), the MMGC asks the MGC of the nearby MGs to dispatch the required load demands from their ESUs. This is achieved by increasing the coordination powers of the nearby MG by a value equal to the load required (i.e., $P_{\mathrm{mg,cd,N_{nmg}}} = P_{\mathrm{mg,cd,N_{nmg}}} + M_{\mathrm{mmg}}(i,5)$ and $Q_{\mathrm{mg,cd,N_{nmg}}} = Q_{\mathrm{mg,cd,N_{nmg}}} + M_{\mathrm{mmg}}(i,6)$ ) and decreasing the coordination powers of the MG under consideration by the same value (i.e., $P_{\mathrm{mg,cd,i}} = P_{\mathrm{mg,cd,i}} - M_{\mathrm{mmg}}(i,5)$ and $Q_{\mathrm{mg,cd,i}} = Q_{\mathrm{mg,cd,i}} - M_{\mathrm{mmg}}(i,6)$ ). If this is not the case, the coordination powers of the nearby MGs are increased by a ratio equal to the power factor of the required load (see Equations (9) and (10)) and the coordination powers of the MG under consideration are decreased by the same value (see Equations (11) and (12)).

$$P_{\mathrm{mg,cd,N_{nmg}}} = P_{\mathrm{mg,cd,N_{nmg}}} + M_{\mathrm{mmg}}(N_{\mathrm{nmg}},3) \times \sqrt{1 + (M_{\mathrm{mmg}}(i,6)/M_{\mathrm{mmg}}(i,5))^2} \tag{9}$$

$$Q_{\mathrm{mg,cd,N_{nmg}}} = Q_{\mathrm{mg,cd,N_{nmg}}} + M_{\mathrm{mmg}}(N_{\mathrm{nmg}},3) \times \sqrt{1 + (M_{\mathrm{mmg}}(i,5)/M_{\mathrm{mmg}}(i,6))^2} \tag{10}$$

$$P_{\mathrm{mg,cd,i}} = P_{\mathrm{mg,cd,i}} - S_{\mathrm{ess,avb,mmg}} \times \sqrt{1 + (M_{\mathrm{mmg}}(i,6)/M_{\mathrm{mmg}}(i,5))^2} \tag{11}$$

$$Q_{\mathrm{mg,cd,i}} = Q_{\mathrm{mg,cd,i}} - S_{\mathrm{ess,avb,mmg}} \times \sqrt{1 + (M_{\mathrm{mmg}}(i,5)/M_{\mathrm{mmg}}(i,6))^2} \tag{12}$$

In this way, all the the contributing MGs supply power with a uniform ratio equal to the power factor of the required load.

- If the load demand at MMG level is higher than the potential of the RESs, EVs, and ESUs, the MMGC deploys the NRESs available for inter-MG operation for the rest of the load supplies. If the NRESs are able to meet the load demands (i.e., $S_{\mathrm{nres,avb,mmg}} \geq \sqrt{(M_{\mathrm{mmg}}(i,5))^2 + (M_{\mathrm{mmg}}(i,6))^2}$), the MGCs of the nearby MGs are dispatched in order of their geographical proximity. This is achieved by increasing the coordination powers of the nearby MG by a value equal to the load required (i.e., $P_{\mathrm{mg,cd,N_{nmg}}} = P_{\mathrm{mg,cd,N_{nmg}}} + M_{\mathrm{mmg}}(i,5)$ and $Q_{\mathrm{mg,cd,N_{nmg}}} = Q_{\mathrm{mg,cd,N_{nmg}}} + M_{\mathrm{mmg}}(i,6)$) and decreasing

the coordination powers of the MG under consideration by the same value (i.e., $P_{\text{mg,cd,i}} = P_{\text{mg,cd,i}} - M_{\text{mmg}}(i, 5)$ and $Q_{\text{mg,cd,i}} = Q_{\text{mg,cd,i}} - M_{\text{mmg}}(i, 6)$ ). If this is not the case, the coordination powers of the nearby MGs are increased by a ratio equal to the power factor of the required load (see Equation (13) and Equation (14)) and the coordination powers of the MG under consideration are decreased by the same value (see Equation (15) and Equation (16)).

$$P_{\text{mg,cd},N_{\text{nmg}}} = P_{\text{mg,cd},N_{\text{nmg}}} + M_{\text{mmg}}(N_{\text{nmg}}, 4) \times \sqrt{1 + (M_{\text{mmg}}(i, 6) / M_{\text{mmg}}(i, 5))^2} \tag{13}$$

$$Q_{\text{mg,cd},N_{\text{nmg}}} = Q_{\text{mg,cd},N_{\text{nmg}}} + M_{\text{mmg}}(N_{\text{nmg}}, 4) \times \sqrt{1 + (M_{\text{mmg}}(i, 5) / M_{\text{mmg}}(i, 6))^2} \tag{14}$$

$$P_{\text{mg,cd,i}} = P_{\text{mg,cd,i}} - M_{\text{mmg}}(N_{\text{nmg}}, 4) \times \sqrt{1 + (M_{\text{mmg}}(i, 6) / M_{\text{mmg}}(i, 5))^2} \tag{15}$$

$$Q_{\text{mg,cd,i}} = Q_{\text{mg,cd,i}} - M_{\text{mmg}}(N_{\text{nmg}}, 4) \times \sqrt{1 + (M_{\text{mmg}}(i, 5) / M_{\text{mmg}}(i, 6))^2} \tag{16}$$

If this is insufficient, the NRESs are dispatched to their full capacity and the MGC of the concerned MG is compelled to shed the remaining loads.

### 2.2. Microgrid Controller (MGC)

The MGC is at the third level of the hierarchy in the proposed control system. It takes information about the available and required powers from the UCs and coordination powers from the MMGC to control the attached UCs accordingly. There are two modes of operation: grid-tied and islanded. The following scenarios are present in the grid-tied mode (see Algorithm 2).

- If there is only active power demand, the RESs are operated in maximum power point tracking (MMPT) mode and the ESUs are charged at their rated capacities, if required. This is perormed to make possible the maximum possible utilization of the RESs and to provide the maximum potential to a particular MG, to meet the load demands by charging ESUs to the maximum possible extent;
- If reactive power is required, it is provided by the RESs, ESUs, and NRESs in order. The rest of the reactive power, if required, is provided by the grid. After the reactive power is scheduled, the RESC extracts the leftover power from the RESs. This is calculated by Equation (17).

$$P_{\text{res,req}} = \sqrt{S_{\text{res,avb}}^2 - Q_{\text{res,req}}^2} \tag{17}$$

The ESUs are charged if there is remaining capacity. If $Q_{\text{ess,d,req}}$ is greater than $S_{\text{ess,do,avb}}$, this means that some portion of $S_{\text{ess,cd,avb}}$ is used to satisfy the load. Thus, the remaining capacity available for charging is given by Equation (18).

$$P_{\text{ess,c,req}} = S_{\text{ess,co,avb}} + \sqrt{S_{\text{ess,cd,avb}}^2 - (Q_{\text{ess,d,req}} - S_{\text{ess,d,o}})^2} \tag{18}$$

However, if $S_{\text{ess,do,avb}}$ is sufficient to satisfy $Q_{\text{ess,d,req}}$, the remaining capacity available for charging is given by Equation (19).

$$P_{\text{ess,c,req}} = S_{\text{ess,co,avb}} + S_{\text{ess,cd,avb}} \tag{19}$$

- If the reactive power is available at MG level, it is used to charge the ESUs of the MG. The rest of the reactive power, if available, is supplied to the grid. The RESs are operated at MPPT. If $Q_{ess,c,req}$ is greater than $S_{ess,co,avb}$, the ESUs are charged to a capacity using Equation (20).

$$P_{ess,c,req} = \sqrt{S_{ess,cd,avb}^2 - (Q_{es,c,req} - S_{ess,co,avb})^2} \qquad (20)$$

However, if $Q_{ess,c,req}$ is smaller than $S_{ess,co,avb}$, the ESUs are charged to a capacity using Equation (21).

$$P_{ess,c,req} = S_{ess,cd,avb} + \sqrt{(S_{ess,co,avb})^2 - (Q_{ess,c,req})^2} \qquad (21)$$

---

**Algorithm 2** Proposed Algorithm for MGC

---

**if** grid-connected **then**
  **if** $Q_{l,req} = 0$ **then**
    Operate RESs at MPPT and charge ESUs at their rated capacity if possible.
  **else if** $Q_{l,req} > 0$ **then**
    Operate RESs, ESUs and NRESs in order if required to supply $Q_{l,req}$. Rest of $Q_{l,req}$, if any, will be provided by the grid.
    Extract the available power from RESs.
    Charge ESUs if possible.
  **else if** $Q_{l,req} < 0$ **then**
    Charge ESUs to the available capacity using grid and available reactive power while delivering rest of it, if any, to the grid.
  **end if**
**else**
  **if** $Q_{l,req} = 0$ **then**
    **if** $P_{ev} = 0$ **then**
      **if** $S_{res,avb} \geq P_{l,req}$ **then**
        Operate RESs to supply the load first and then ESUs if possible. Curtail RESs if required.
      **else**
        Operate RESs, ESUs and NRESs in order if required. Shed the load if required.
      **end if**
    **else if** $P_{ev} > 0$ **then**
      **if** $S_{res,avb} \geq P_{l,req}$ **then**
        Operate RESs to supply $P_{l,req}$.
        **if** $S_{res,avb} > P_{ev}$ **then**
          Operate RESs to supply $P_{ev}$.
          **if** $S_{res,avb} > S_{ess,c,avb}$ **then**
            Operate RESs to supply $S_{ess,c,avb}$ and curtail the rest.
          **else**
            Operate the remaining capacity of RESs to charge ESUs.
          **end if**
        **else**
          Operate RESs at MPPT
          **if** $S_{ess,d,avb} \geq P_{ev,d}$ **then**
            Operate ESUs to supply $P_{ev}$ completely.
          **else**
            Operate ESUs to supply the appropriate portion of $P_{ev}$.
            **if** $S_{nres,avb} > P_{ev,d}$ **then**
              Operate NRESS to supply $P_{ev}$ completely.
            **else**
              Operate NRESs to supply the appropriate portion of $P_{ev}$ and shed the rest.
            **end if**
          **end if**
        **end if**
      **else**
        Operate RESs at MPPT.
        **if** $S_{ess,d,avb} \geq P_{l,d}$ **then**
          Use ESUs to supply the load completely.
          **if** $S_{ess,d,avb} \geq P_{ev}$ **then**
            Use ESUs to supply EVs completely.
          **else**
            Operate EVs at the full capacity.
            **if** $S_{nres,avb} > P_{ev,d}$ **then**
              Operate NRESs to supply the remaining EV load.
            **else**
              Operate NRESs at full capacity and shed the remaining EV load.
            **end if**
          **end if**
        **else**
          Use ESUs fully to supply the load.
          **if** $S_{nres,avb} > P_{l,d}$ **then**

          Operate NRESs to supply the load first and then EVs if possible. Shed EV demand accordingly if required.
        **else**
           Operate NRESs at full capacity. Shed the remaining load and entire EV demand.
        **end if**
      **end if**
    **end if**
  **else**
    **if** $S_{res,avb} \geq P_{l,req}$ **then**
      Operate RESs to supply $P_{l,req}$
      **if** $S_{res,avb} > S_{ess,c,avb}$ **then**
        Operate RESs to supply $S_{ess,c,avb}$ and curtail the rest.
        Curtail $P_{ev,avb}$.
      **else**
        Operate RESs at MPPT
        **if** $P_{ev,avb} \geq S_{ess,c,avb}$ **then**
          Use $P_{ev,avb}$ to supply $S_{ess,c,avb}$ and curtail the rest.
        **else**
          Use $P_{ev,avb}$ completely to charge ESUs
        **end if**
      **end if**
    **else**
      Operate RESs at MPPT
      **if** $P_{ev,avb} \geq P_{l,d}$ **then**
        Use $P_{ev,avb}$ to supply $P_{l,d}$
        **if** $P_{ev,avb} > S_{ess,c,avb}$ **then**
          Use $P_{ev,avb}$ to supply $S_{ess,c,avb}$ completely
        **else**
          Use $P_{ev,avb}$ to supply a portion of $S_{ess,c,avb}$
        **end if**
      **else**
        Use $P_{ev,avb}$ to supply a portion of $P_{l,d}$
        **if** $S_{ess,d,avb} \geq P_{l,d}$ **then**
          Use $S_{ess,d,avb}$ to supply $P_{l,d}$
        **else**
          Operate ESUs at their full capacity
          **if** $S_{nres,avb} > P_{l,d}$ **then**
            Use NRESs to supply the load
          **else**
            Operate NRESs at their full capacity and shed the remaining load
          **end if**
        **end if**
      **end if**
    **end if**
  **else if** $Q_{l,req} > 0$ **then**
    Operate RESs, ESUs and NRESs in order if required. Rest of the $Q_{l,req}$, if any, will be shedded.
    Calculate $P_{res,avb}$, $P_{ess,c,avb}$, $P_{ess,d,avb}$ and $P_{nres,avb}$.
    Perform the rest of the active power scheduling as in case of $Q_{l,req=0}$
  **else if** $Q_{l,req} < 0$ **then**
    Charge ESUs using the available reactive power while shedding rest of it, if any.
    Calculate $P_{res,avb}$, $P_{ess,c,avb}$, $P_{ess,d,avb}$ and $P_{nres,avb}$
    Perform the rest of the active power scheduling as in case of $Q_{l,req=0}$
  **end if**
**end if**
Calculate $S_{res,avb,mmg}$, $P_{ev,avb,mmg}$, $S_{ess,d,avb,mmg}$, $S_{nres,avb,mmg}$, $P_{mg,req,mmg}$, $Q_{mg,req,mmg}$ for MMGC

---

In islanded mode, the MGC is responsible for supplying a load to the maximum extent from the available resources. If the available resources are insufficient, the required load is shed.

- If there is only active power demand and the EVs are unable to load or support the MG, the RESs are considered for supplying the load at first. If the RESs are sufficient for load supply, extra power from the RESs is used to charge the ESUs. The rest of the RES power, if available, is curtailed. Otherwise, the ESUs and NRESs are used to supply the load, in order, if needed. The rest of the load, if any, is shed.
- In the presence of only an active power load and EV charging demand, the RESs are used to supply the load first. If the RESs are sufficient for load supply, the remaining RES capacity is used to supply the EVs. The remaining RES capacity, if any, is used to charge the ESUs if possible. If the RESs, after supplying the load, are not able to supply the EVs, the ESUs and NRESs are used to supply the EVs, in order, if required. If the RESs are not able to supply the load, the ESUs are used to supply the remaining load. If the ESUs are able to supply the remaining load, the EVs are supplied from the ESUs. If the ESUs are insufficient for the supply, the NRESs are deployed accordingly. In this case, even the NRESs fall short of the EV demand and EVs are shed accordingly.If the both RESs and ESUs are insufficient to fulfill the load demand, the RESs and ESUs are fully utilized and the remaining load demand is fulfilled from the NRESs. If the NRESs are sufficient to fulfill the load demand, the rest of the NRES capacity is used to supply the EVs. If this capacity is insufficient, the remaining EV load is shed. If the NRESs are not even sufficient to fulfill the load demand, the NRESs are operated at their rated capacity. The remaining load and the whole of the EV demand is shed.
- If the load is purely active and the EVs are able to supply the load, the RESs are used to supply the load at first. The ESUs are charged from the remaining RES potential and, if required, from the power available from the EVs. If the RESs are not sufficient to supply the load, the RESs are fully utilized and the remaining load is supplied from the EVs. The remaining EV capacity, if any, is used to charge the ESUs, if required. Any more EV capacity, if available, is curtailed. If both RESs and EVs are insufficient, the ESUs and NRESs are used to supply the remaining load.
- In the case of a reactive power requirement, this is supplied from the RESs, ESUs, and NRESs, and the remaining reactive power, if any, is shed. After that $P_{\text{res,avb}}$, $P_{\text{ess,c,avb}}$, $P_{\text{ess,d,avb}}$, and $P_{\text{nres,avb}}$ are calculated for active power dispatch. $P_{\text{res,avb}}$ can be calculated using Equation (22).

$$P_{\text{res,avb}} = \sqrt{S_{\text{res,avb}}^2 - Q_{\text{res,req}}^2} \tag{22}$$

If $Q_{\text{ess,d,req}} > S_{\text{ess,do,avb}}$, this means that some portion of $S_{\text{ess,cd,avb}}$ is discharged to meet the reactive power requirements, so, $P_{\text{ess,c,avb}}$ and $P_{\text{ess,d,avb}}$ can be calculated using Equations (23) and (24).

$$P_{\text{ess,c,avb}} = S_{\text{ess,co,avb}} + \sqrt{S_{\text{ess,cd,avb}}^2 - (Q_{\text{ess,d,req}} - S_{\text{ess,do,avb}})^2} \tag{23}$$

$$P_{\text{ess,d,avb}} = \sqrt{S_{\text{ess,cd,avb}}^2 - (Q_{\text{ess,d,req}} - S_{\text{ess,do,avb}})^2} \tag{24}$$

If $Q_{\text{ess,d,req}} \leq S_{\text{ess,do,avb}}$, this means that some portion of $S_{\text{ess,do,avb}}$ can be used to meet the active power requirements. Thus, $P_{\text{ess,c,avb}}$ and $P_{\text{ess,d,avb}}$ can be calculated using Equations (25) and (26).

$$P_{\text{ess,c,avb}} = S_{\text{ess,co,avb}} + S_{\text{ess,cd,avb}} \tag{25}$$

$$P_{\text{ess,d,avb}} = S_{\text{ess,cd,avb}} + \sqrt{S_{\text{ess,do,avb}}^2 - Q_{\text{ess,d,req}}^2} \tag{26}$$

$P_{\text{nres,avb}}$ for active power dispatch can be calculated using Equation (27).

$$P_{\text{nres,avb}} = \sqrt{S_{\text{nres,avb}}^2 - Q_{\text{nres,req}}^2} \tag{27}$$

After calculating the remaining capacities of the RESs, ESUs, and NRESs, the active power is scheduled as describe above.

- If reactive power is available, it is absorbed by the ESUs, and left over reactive power, if any, is shed. Since RESs are not used in this process, $P_{\text{res,avb}}$ remains intact to meet the active power demands (see Equation (28)).

$$P_{\text{res,avb}} = S_{\text{res,avb}} \tag{28}$$

If $Q_{\text{ess,c,req}} > S_{\text{ess,co,avb}}$, this means that some portion of $S_{\text{ess,cd,avb}}$ is used for reactive power support. Thus, the available resources from ESUs towards active power support can be calculated using Equations (29) and (30).

$$P_{\text{ess,c,avb}} = \sqrt{S^2_{\text{ess,cd,avb}} - (Q_{\text{ess,c,req}} - S_{\text{ess,co,avb}})^2} \tag{29}$$

$$P_{\text{ess,d,avb}} = S_{\text{ess,do,avb}} + \sqrt{S^2_{\text{ess,cd,avb}} - (Q_{\text{ess,c,req}} - S_{\text{ess,co,avb}})^2} \tag{30}$$

If $Q_{\text{ess,c,req}} \leq S_{\text{ess,co,avb}}$, this means that some portion of $S_{\text{ess,cd,avb}}$ can be used for active power support. Thus, the available resources from ESUs towards active power support can be calculated using Equations (31) and (32).

$$P_{\text{ess,c,avb}} = S_{\text{ess,cd,avb}} + \sqrt{(S_{\text{ess,co,avb}})^2 - (Q_{\text{ess,c,req}})^2} \tag{31}$$

$$P_{\text{ess,d,avb}} = S_{\text{ess,cd,avb}} + S_{\text{ess,do,avb}} \tag{32}$$

Since NRESs are not used in this process, $P_{\text{nres,avb}}$ remains intact to meet the active power demands (see Equation (33)).

$$P_{\text{nres,avb}} = S_{\text{nres,avb}} \tag{33}$$

After that, the active power is scheduled as describe above.

### 2.3. EV Controllers (EVCs)

The EVC controls the charging and discharging of the EVs in coordination with the MGC and MMGC. Each vehicle, on arrival, informs its MC about its $C_{\text{ev,i}}$, $P_{\text{r,ev,i}}$, $SoC_{\text{i,ev,i}}$, $SoC_{\text{f,ev,i}}$, $SoC_{\text{min,ev,i}}$, $SoC_{\text{max,ev,i}}$, $cost_{\text{ev,i}}$, $m_{\text{ev,i}}$, $t_{\text{a,ev,i}}$, and $t_{\text{d,ev,i}}$. Based on this information, the respective MC determines the reference power demand of the EV and conveys it to the respective UC. The UC analyzes the power demands of all the EVs in a MG and conveys it to the MGC. The MGC interacts with the MMGC to see if these demands can be met or if there is a need to perform shedding or curtailment (see Algorithm 3).

Each EV can select a mode of charging or discharging. "Minimum time charging mode" means that the EV needs to be charged in the minimum possible time. The minimum charging time can be achieved if the EV is charged at its rated power, i.e., $P_{\text{r,ev,i}}$. First of all, the MC determines the possibility of charging the EV in that time. This can be achieved by comparing the required energy $(SoC_{\text{f,ev,i}} - SoC_{\text{i,ev,i}}) \times C_{\text{ev,i}}$ and the maximum energy that can be made available within the specified time limits $(P_{\text{r,ev,i}} \times (t_{\text{d,ev,i}} - t_{\text{a,ev,i}}))$, i.e.,

$$\frac{(SoC_{\text{f,ev,i}} - SoC_{\text{i,ev,i}}) \times C_{\text{ev,i}}}{P_{\text{r,ev,i}} \times (t_{\text{d,ev,i}} - t_{\text{a,ev,i}})} \tag{34}$$

---

**Algorithm 3** Proposed EVC Algorithm

---

**if** $m_{\text{ev,i}}$ = "Minimum Time Charging Mode" **then**
　Check the feasibility of $SoC_{\text{f,ev,i}}$ and update it if required
　**if** $(t_{\text{c,ev,i}} \geq t_{\text{a,ev,i}})$ & $(t_{\text{c,ev,i}} < t_{\text{d,ev,i}})$ & $(SoC_{\text{c,ev,i}} \leq SoC_{\text{f,ev,i}})$ **then**
　　$P_{\text{ev,i}} = P_{\text{r,ev,i}}$
　**else**
　　$P_{\text{ev,i}} = 0$
　**end if**
**else if** $m_{\text{ev,i}}$ = "Minimum Time Fixed Cost Charging Mode" **then**
　Check the feasibility of $SoC_{\text{f,ev,i}}$ and update it if required
　**if** $(t_{\text{c,ev,i}} \geq t_{\text{a,ev,i}})$ & $(t_{\text{c,ev,i}} < t_{\text{d,ev,i}})$ & $(SoC_{\text{c,ev,i}} \leq SoC_{\text{f,ev,i}})$ & $(cost_{\text{c}} <$
　$cost_{\text{ev,i}}$ **then**
　　$P_{\text{ev,i}} = P_{\text{r,ev,i}}$
　**else**
　　$P_{\text{ev,i}} = 0$
　**end if**
**else if** $m_{\text{ev,i}}$ = "Minimum Cost Charging Mode" **then**
　**if** MG is in grid connected mode **then**
　　Check the feasibility of $SoC_{\text{f,ev,i}}$ and update it if required
　　Solve

$$\underset{P_{\text{ev,i}}}{\text{minimize}} \quad \sum_{i=1}^{N} P_{\text{ev,i}}(i) \times cost_{\text{avb}}(i) \times \Delta T$$

　　subject to

$$SoC_{\text{ev,i}}(1) = SoC_{\text{i,ev,i}}$$

$$SoC_{\text{ev,i}}(2:N-1) = SoC_{\text{ev,i}}(1:N-2) + P_{\text{ev,i}}(1:N-2) \times \Delta T$$

$$SoC_{\text{ev,i}}(N-1) + P_{\text{ev,i}}(N) \times \Delta T = SoC_{\text{f,ev,i}}$$

　**else**
　　$P_{\text{ev,i}} = 0$
　**end if**
**else if** $m_{\text{ev,i}}$ = "Fixed Cost Maximum SoC Charging Mode" **then**
　**if** MG is in grid connected mode **then**
　　Check the feasibility of $SoC_{\text{f,ev,i}}$ and update it if required
　　Solve

$$\underset{P_{\text{ev,i}}}{\text{maximize}} \quad SoC_{\text{i,ev,i}}(i) + \frac{\sum_{i=1}^{N} P_{\text{ev,i}}(i) \times \Delta T}{C_{\text{ev,i}}}$$

　　subject to

$$\sum_{i=1}^{N} P_{\text{ev,i}}(i)\Delta T \times cost_{\text{avb}}(i) \times \Delta T \leq cost_{\text{ev,i}}$$

$$SoC(1) = SoC_i$$

$$SoC(2:N) = \frac{SoC(1:N-1) + P_{\text{ev,i}}(1:N-1) \times \Delta T}{C_{\text{ev,i}}}$$

　**else**
　　$P_{\text{ev,i}} = 0$
　**end if**
**else if** $m_{\text{ev,i}}$ = "Grid Support Mode" **then**
　Check the feasibility of $SoC_{\text{f,ev,i}}$ and update it if required
　Solve

$$\underset{P_{\text{ev,i}}}{\text{maximize}} \quad \sum_{i=1}^{N} \Delta T \times cost_{\text{avb}}(i) \times P_{\text{ev,i}}(i)$$

　subject to

$$SoC_{\text{ev,i}}(1) = SoC_{\text{i,ev,i}}$$

$$SoC_{\text{ev,i}}(1:i_a-2) + P_{\text{ev,i}}(1:i_a-2) \times \Delta T = SoC_{\text{i,ev,i}}$$

$$SoC_{\text{ev,i}}(i_a:i_d-1) = SoC_{\text{ev,i}}(i_a-1:i_d-2) + P_{\text{ev,i}}(i_a-1:i_d-2) \times \Delta T$$

$$SoC_{\text{ev,i}}(i_d-1:N-1) + P_{\text{ev,i}}(i_d-1:N-1) \times \Delta T = SoC_{\text{f,ev,i}}$$

$$P_{\text{ev,i}}(1:i_a) = 0$$

$$P_{\text{ev,i}}(i_d+1:N) = 0$$

**end if**

---

If Equation (34) is greater than 1, the required energy cannot be fulfilled in the specified time. In such a case, $SoC_{\text{f,ev,i}}$ can be reduced to a value that can be achieved within the specified time duration while charging at the maximum possible power. This can be calculated using Equation (35).

$$SoC_{\text{f,ev,i}} = SoC_{\text{i,ev,i}} + \frac{P_{\text{r,ev,i}} \times (t_{\text{d,ev,i}} - t_{\text{a,ev,i}})}{C_{\text{ev,i}}} \tag{35}$$

If Equation (34) is less than 1, the required energy can be supplied within the specified time. In such a case, $SoC_{\text{f,ev,i}}$ is kept as it is. After that, the EV is charged at its rated power until $t_{\text{d,ev,i}}$ is reached or $SoC_{\text{c,ev,i}}$ becomes equal to $SoC_{\text{f,ev,i}}$.

"Minimum time fixed cost charging mode" means that the EV is needs to be charged in the minimum possible time, with the constraint that the cost should not exceed the cost specified by the EV. First of all, the MC determines whether the EV can be charged to its $SoC_{\text{f,ev,i}}$ within the specified time duration. If this is not possible, $SoC_{\text{f,ev,i}}$ is updated accordingly, otherwise it is kept as such. After that, the EV is charged at its rated power until $t_{\text{d,ev,i}}$ is reached or $SoC_{\text{c,ev,i}}$ becomes equal to $SoC_{\text{f,ev,i}}$ or the current cost of EV ($cost_{\text{c,ev,i}}$) is equal to the specified cost ($cost_{\text{sp,ev,i}}$).

"Minimum cost charging mode" means that the EV needs to be charged with the minimum possible cost to a specific $SoC_{\text{f,ev,i}}$ within a given time period. First of all, the MC determines the feasibility of $SoC_{\text{f,ev,i}}$ and updates it if required, as explained above. After that, an optimization problem is solved to minimize the cost in such a way that $SoC_{\text{f,ev,i}}$ is met within a given time duration by keeping $P_{\text{ev,i}}$ within the limits, as given by Equation (36).

$$\underset{P_{ev,i}}{\text{minimize}} \quad \sum_{i=1}^{N} P_{ev,i}(i) \times cost_{avb}(i) \times \Delta T$$

subject to

$$SoC_{ev,i}(1) = SoC_{i,ev,i}$$

$$SoC_{ev,i}(2:N-1) = SoC_{ev,i}(1:N-2) + P_{ev,i}(1:N-2) \times \Delta T \qquad (36)$$

$$SoC_{ev,i}(N-1) + P_{ev,i}(N) \times \Delta T = SoC_{f,ev,i}$$

Here, $i$ represents the optimization interval, which ranges from 1 to $N$, and $cost_{avb}(i)$ represents the cost of charging the EV at the instant $i$. These calculations remain valid until the grid is connected or the MGC does not shed any demand. During such events, the EV is not charged at the available power, and the calculations are performed again when such events are over.

"Maximum SoC fixed cost charging mode" means that the EV needs to be charged to achieve the maximum possible $SoC_{f,ev,i}$, while keeping the cost within a specified cost. First of all, the MC determines the feasibility of $SoC_{f,ev,i}$ and updates it if required, as explained above. After that, an optimization problem is solved to maximize the SoC at each charging interval, while meeting the time, cost, and power constraints as given by Equation (37).

$$\underset{P_{ev,i}}{\text{maximize}} \quad SoC_{i,ev,i}(i) + \frac{\sum_{i=1}^{N} P_{ev,i}(i) \times \Delta T}{C_{ev,i}}$$

subject to

$$\sum_{i=1}^{N} P_{ev,i}(i)\Delta T \times cost_{avb}(i) \times \Delta T \leq cost_{ev,i} \qquad (37)$$

$$SoC(1) = SoC_i$$

$$SoC(2:N) = \frac{SoC(1:N-1) + P_{ev,i}(1:N-1) \times \Delta T}{C_{ev,i}}$$

"Grid support mode" means that the EV needs to be charged or discharged to achieve a specified SoC within a given time interval, while maintaining the EV constraints of power with the maximum possible cost benefit. First of all, the MC determines whether the EV can be charged to its $SoC_{f,ev,i}$ within the specified time duration. If this is not possible, $SoC_{f,ev,i}$ is updated accordingly, otherwise it is kept as such, as explained above. After that, an optimization problem is formulated to charge or discharge the EV, so that $SoC_{f,ev,i}$ is achieved at the end of the specified time and in such a way that the maximum cost benefit is achieved. Such a problem is stated in Equation (38).

$$\underset{P_{ev,i}}{\text{maximize}} \quad \sum_{i=1}^{N} \Delta T \times cost_{avb}(i) \times P_{ev,i}(i)$$

subjet to

$$SoC_{ev,i}(1) = SoC_{i,ev,i}$$

$$SoC_{ev,i}(1:i_a-2) + P_{ev,i}(1:i_a-2) \times \Delta T = SoC_{i,ev,i}$$

$$SoC_{ev,i}(i_a:i_d-1) = SoC_{ev,i}(i_a-1:i_d-2) + P_{ev,i}(i_a-1:i_d-2) \times \Delta T \qquad (38)$$

$$SoC_{ev,i}(i_d-1:N-1) + P_{ev,i}(i_d-1:N-1) \times \Delta T = SoC_{f,ev,i}$$

$$P_{ev,i}(1:i_a) = 0$$

$$P_{ev,i}(i_d+1:N) = 0$$

It is to be noted that any suitable algorithm can be used to control RESs, ESUs, NRESs and loads. A particular example can be found in [60].

## 3. Results and Discussion

A comprehensive case study based on five MGs was developed to validate the behavior of the MG controllers. Each MG had solar panels as RESs, BESS as ESUs, and EVs and diesel generators as NRESs. First, the fourth and fifth MGs were operated at unity power factors, the second MG was operated at lagging power factor, and the third MG was operated at leading power factor. In this way, the behavior of MG controllers for the active and reactive (leading and lagging) conditions could be verified. All the MGs had the same solar profile, except the fifth MG, which had a higher solar profile. The fourth MG had a higher load and a lower solar profile than the fifth MG and, therefore, required MMG interaction. This provided a way to validate the behavior of the MG controllers at MMG level. Symmetry was kept when selecting the parameters of the different DESs, in order to ease the understanding of the behavior of the different controllers in such a complex MMG environment. This will become evident from the following discussion. The following sections present a detailed discussion on the behavior of the different controllers at different levels of the hierarchy for all MGs involved.

### 3.1. Response of MG 1

Figure 3 represents the behavior of the different controllers for the first MG. Figure 3a represents the resources of MG 1 that were able to participate in MMG interaction. The solar resource was not available to participate at MMG level during the night hours; thus, a zero line can be seen for $S_{res,avb,mmg,1}$ in Figure 3a from 0 to 5 and 19 to 24 h. In between these hours, there was availability of the solar resource. In grid-connected mode, the whole solar resource was available at the MMG level, showing that the grid was responsible for supplying the load and the solar resource could be used to supply the other MGs if needed. In islanded mode, the solar resource of the MG was configured to supply the load. Thus, we saw a reduction in solar availability during these hours. During the grid-connected hours, the ESUs were free to participate in MMG interaction; therefore, the whole of the discharge capacity ($S_{ess,do,avb,mg,i} + S_{ess,cd,avb,mg,i}$) was available at MMG level. In islanded mode, the ESUs were engaged in supplying the load after using RESs and EVs. As a result, their participation at MMG level was reduced in islanded mode. The NRESs were not deployed in grid-connected mode; thus, the full capacity of the NRESs of the MG was available for MMG interaction. In the case of the islanded modes, the NRESs were deployed if other sources were not sufficient to supply the load. Thus, the availability of NRESs at MMG level was thereby reduced (see Figure 3d) during these hours. It can be observed that it was a well balanced MG, as it did not require any power from the MMGC, as represented by zero lines for $P_{mg,req,1}$, $Q_{mg,req,1}$, $P_{mg,cd,1}$, and $Q_{mg,cd,1}$ in Figure 3a.

Figure 3b represents the behavior of the MGC. The MGC accepted the available solar capacity at the MG level from the RESC, charging and discharging capacity from the ESUs by means of the ESSC, available and required power from the EVs by means of EVCs, rated capacity of the NRESs from NRESC, and the required load from the LC. Having the information about the required load and available resources, the MGC makes decisions for each controller at the unit level, to provide or shed a certain load and curtail a certain source. The concerned MG operates at the unity power factor; thus, we did not see any reactive power scheduling here. RESs were operated at their full capacity, with only one exception where a curtailment took place during a low load, and the ESUs were fully charged and no grid was present. The ESUs were controlled to be charged as much as possible from the RESs, EVs, and grid in order, if possible, and discharged only when required. The same trend can be observed in Figure 3b. The NRESs were kept idle in grid-connected mode and only operated during islanding when other sources could not support the load. The load was fully supplied in grid-connected mode and load shedding was only performed in islanded mode when the available MG resources were not sufficient to supply the load.

Figure 3c represents the behavior of the RESC at the MG level. It can be seen that the RESs were extracted to the maximum extent in grid-connected mode. In islanded mode,

the RESs were curtailed if these were in excess, even after supplying the loads, ESUs, and EVs, as happened from 14 to 15 h.

Figure 3d represents the behavior of the NRESC at MG level. The NRESs were not operated in grid-connected mode, where the grid maintains the load supply, while accepting RESs at MPPT and charging the ESUs at their available capacity. The NRESs were operated in islanded mode as a last resort when the other available resources were not sufficient to supply the loads.

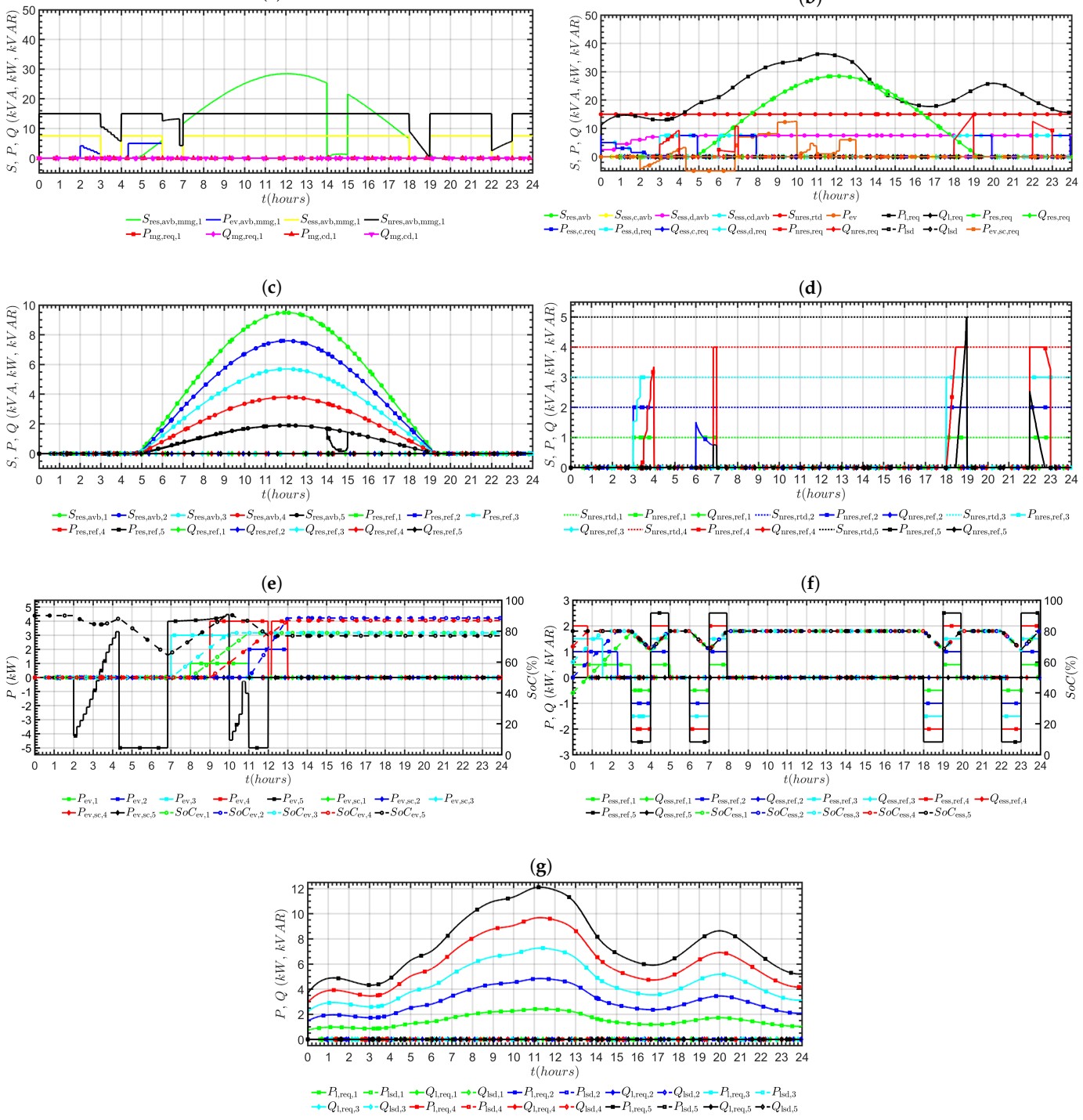

**Figure 3.** Behavior of MG 1 controllers. (**a**) MMGC for MG 1; (**b**) MGC 1; (**c**) RESC 1; (**d**) NRESC 1; (**e**) EVC 1; (**f**) ESSC 1; (**g**) LC 1.

Figure 3f represents the behavior of the ESSC at MG level. The ESUs were charged whenever the grid was available or the RESs were sufficient to supply the load, and they were only discharged when the grid was not available and the RESs were not sufficient. The same trend was observed here. During the hours of solar deficiency and islanded mode, the ESUs were discharged to supply the load and charged whenever the grid was connected after that.

Figure 3e represents the behavior of the EVC at MG level. The first EV chose "minimum time charging mode", the second EV chose "minimum time fixed cost charging mode", the third EV chose "minimum cost charging mode", the fourth EV chose "fixed cost maximum SoC charging mode", and the fifth EV chose "grid-support mode". Since the MG was fairly self sufficient, there was no power shedding as far as the EV demand was concerned.

Figure 3g represents the behavior of the load in the first MG. It can be observed that the load was always supplied during grid-connected mode, and load shedding was observed in islanded mode when no available resources could sufficiently meet the load requirements.

*3.2. Response of MG 2*

Figure 4 represents the behavior of the different controllers for the second MG. Figure 4a represents the resources of the MG 2 that were able to participate in MMG interaction. It can be seen that the MG was quite self-sufficient. In the grid-tied mode of operation, there were resources available that could participate in the MMG operation. However, in such a case, the grid took care of the load supply imbalance. However, such resources could be used in the case of partial islanding in some other MGs. Thus, we can see that $P_{mg,cd,2}$ and $Q_{mg,cd,2}$ were zero in such a mode. In the islanded mode of operation, the MG was able to provide its loads but it did not have enough resources to participate in MMG operation. That was reflected by zero values of $S_{res,avb,mmg,2}$, $P_{ev,avb,mmg,2}$, $S_{ess,avb,mmg,2}$, $S_{nres,avb,mmg,2}$, $P_{mg,req,2}$, $Q_{mg,req,2}$, $P_{mg,cd,2}$, and $Q_{mg,cd,2}$ in the islanded mode of operation.

Figure 4b represents the behavior of the MGC for the second MG. The MGC accepted the available and required capacities at the MG level from all the unit controllers and took decisions for each controller at the unit level, to provide or shed a certain load and curtail a certain source. The MG operated at the lagging power factor. Thus, we saw both active and reactive powers here. During the hours of grid-tied mode operation, the MGC showed full utilization of $S_{res,avb}$ by operating RESs at MPPT. In islanded mode of operation, the RESs were operated to meet the requirements of the load required. It can be observed that from 14 to 15 h, the load demands were low compared to the available capacity of the RESs. Thus, we saw a curtailment of the RESs. The ESUs were charged in the grid-tied mode of operation if required. In the islanded mode of operation, the ESUs were charged if enough RESs and EVs were available beyond the requirements of the loads. The NRESs were kept idle in the grid-tied mode of operation and operated only if required. Since the reactive power demand could be compensated for by the resources of the MG, we can see that there was no reactive power load shedding.

Figure 4c represents the behavior of the RESC at the MG level. The RESC took the active and reactive power commands from the MGC and distributed them to the respective RESs according to the control algorithm, to meet the requirements of the MGC. Since we had reactive power demands as well, the respective MC controlled the power electronic interface to meet both the active and reactive power demands according to the requirements of the MGC.

Figure 4d represents the behavior of the NRESC at MG level. It can be seen that the NRESs were not operated in grid-tie mode, as the grid was responsible for maintaining the load supply balance. On the other hand, the NRESs were operated during islanding, when the other DESs were not sufficient to meet the load requirements, as occurred from 3 to 4, 6 to 7, 18 to 19, and 22 to 23 h. In the islanded mode of operation from 14 to 15 h, the RESs were available above the load demand. Thus, we observed a curtailment of RESs, and

hence the NRESs were not operated, as this was not required as the load had already been satisfied.

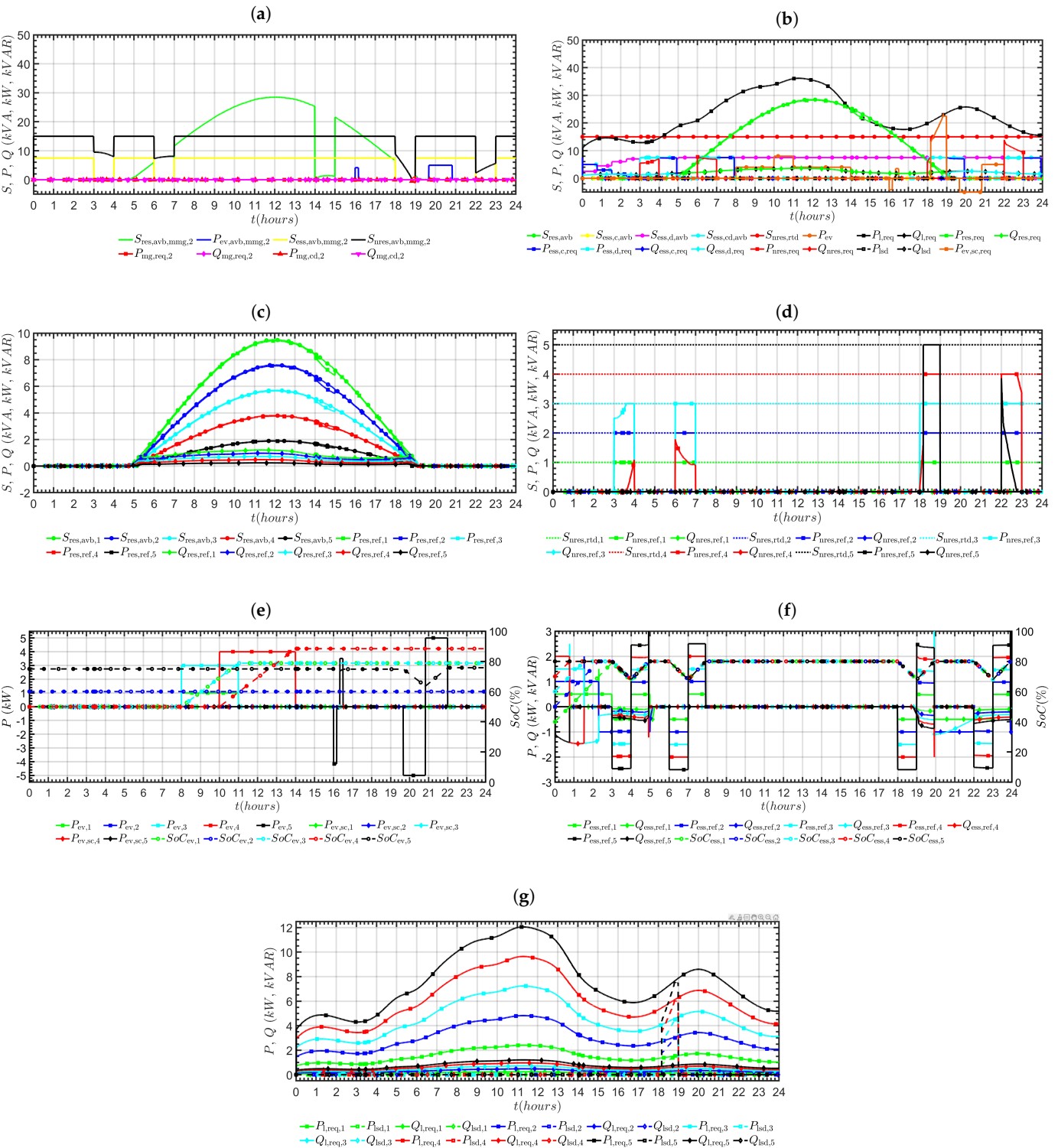

**Figure 4.** Behavior of MG 2 controllers. (**a**) MMGC for MG 2; (**b**) MGC 2; (**c**) RESC 2; (**d**) NRESC 2; (**e**) EVC 2; (**f**) ESSC 2; (**g**) LC 2.

Figure 4f represents the behavior of the ESSC at MG level. ESUs are charged whenever the grid was available or RESs are sufficient to supply the load, and discharged only when the grid was not available and the RESs were not sufficient. The same trend was observed

here. During the hours of solar deficiency and islanded mode, the ESUs were discharged to supply the load and charged whenever the grid was connected after that. It can be observed that ESUs were discharged only during islanding when the RESs were not sufficient to supply the load. For example, from 14 to 15 h, the RESs were curtailed. This means that the RESs could supply the load, and hence the ESUs were not discharged, even in the absence of the grid.

Figure 4e represents the behavior of the EVC at MG level. The first EV chose the "minimum time charging mode", the second EV chose "minimum time fixed cost charging mode", the third EV chose "minimum cost charging mode", the fourth EV chose "fixed cost maximum SoC charging mode", and the fifth EV chose "grid-support mode". All the vehicles operated according to their operating modes.

Figure 4g represents the behavior of the LC in the second MG. It can be observed that the load was always supplied in grid-connected mode and load shedding was observed in islanded mode when the available resources could not sufficiently meet the load requirements. There was no reactive power load shedding, since the MG tried to preferably compensate for the reactive power from its own resources. That is why we saw active power load shedding during islanding conditions from 18 to 19 h when the RESs were not present and the NRESs and ESUs were insufficient to supply the load even when operated at full capacity. The MG required $P_{\mathrm{mg,req,2}}$ from the MMGC, as can be seen in Figure 4a, but there was no $P_{\mathrm{mg,cd,2}}$ corresponding to the request, as there was no nearby available MG to meet this demand.

### 3.3. Response of MG 3

Figure 5 represents the behavior of the different controllers for the third MG. Figure 5a represents the resources of MG 3 that was able to participate in the MMG interaction. It can be seen that the RESs were available at MMG level during grid-tied mode, as the grid took care of all the feed from the RESs. Since the MG had a leading power factor, the ESUs were used to absorb this reactive power; thus, $S_{\mathrm{ess,avb,mmg,3}}$ was smaller compared to the corresponding MG with unity power factor. The NRESs were idle in grid-connected mode; therefore, full capacity was available to participate in the MMG interaction. It can be seen that the MG was quite well self balanced. Thus, there was no $P_{\mathrm{mg,req,3}}$ and $Q_{\mathrm{mg,req,3}}$. It can also be seen that the MG resources were not in excess of those needed in islanded mode; thus, there was also no $P_{\mathrm{mg,cd,3}}$ and $Q_{\mathrm{mg,cd,3}}$. Figure 5b represents the behavior of MGC for the third MG. It can be observed that the MG resources were sufficient for the MG requirements.

Figure 5c represents the behavior of the RESC at MG level. Since the MG was operating at a leading power factor, the stress on the ESUs was higher in islanded mode compared to the second MG. Therefore, we can see that the RESs were controlled to operate at their full potential. A curtailment was observed from 14 to 15 h, representing an excess of RESs. Figure 5d represents the behavior of the NRESC at MG level. The NRESs were kept idle during the grid-tied mode and only operated during islanding, when the other DESs were not able to support the load. The same behavior was observed here. Figure 5e represents the behavior of the ESSC at MG level. Since the MG operated at the leading power factor, we can observe that the ESUs were charged when there was an excess of reactive power. Figure 5f represents the behavior of the EVC at MG level. The first EV chose "minimum time charging mode", the second EV chose "minimum time fixed cost charging mode", the third EV chose "minimum cost charging mode", the fourth EV chose "fixed cost maximum SoC charging mode", and the fifth EV chose "grid-support mode". The operating modes were followed by the coordination of MGC and MMGC. Figure 5g represents the behavior of the LC. Since the MG had enough resources, there was no load shedding.

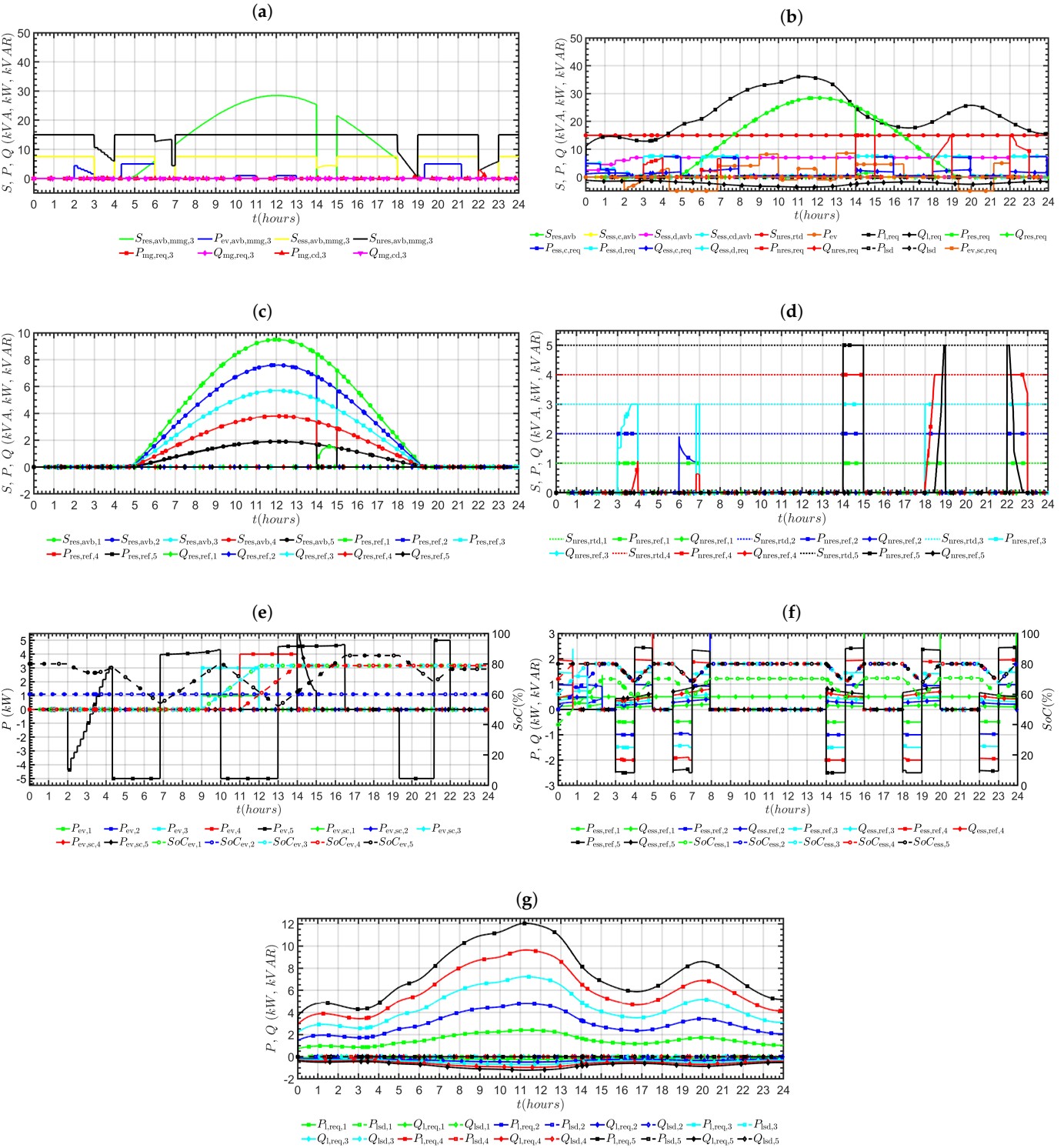

**Figure 5.** Behavior of MG 3 controllers. (**a**) MMGC for MG 3; (**b**) MGC 3; (**c**) RESC 3; (**d**) NRESC 3; (**e**) EVC 3; (**f**) ESSC 3; (**g**) LC 3.

### 3.4. Response of MG 4

Figure 6 represents the behavior of the different controllers for the fourth MG. This MG was identical to the first MG, with the exception of a high load. Thus, we can see a negative coordination power during the islanding instants from 18 to 19 and 22 to 23 h (see Figure 6a). This means that the MG was not able to meet its load from the available

resources and requested support from the MMGC. The MMGC used the geographically nearby MGs that were able to support this MG (see Figures 5a and 7a) to meet the MG demand at MMG level. Figure 6b represents the behavior of the MGC for this MG. The MGC fulfilled the load demands from the available resources and requested support from the other MGs through the MMGC when the available resources fell short of providing the MG load demands. Figure 6c represents the behavior of the RESC at the MG level. It can be observed that the RESs were fully utilized. Figure 6d represents the behavior of the NRESC at MG level. The NRESs were kept idle in grid-tie mode, while they operated in islanded mode only if all the other sources were not able to meet the MG demands. Figure 6f represents the behavior of the ESSC at MG level. Since the MG had a high load, the ESUs were always discharged in islanded mode. This is in contrast to the behavior of similar MGs with lower load demands. Figure 6e represents the behavior of the EVC at MG level. The first EV chose "minimum time charging mode", the second EV chose "minimum time fixed cost charging mode", the third EV chose "minimum cost charging mode", the fourth EV chose "fixed cost maximum SoC charging mode", and the fifth EV chose "grid-support mode". All the vehicles followed their operating modes. Figure 6g represents the behavior of the LC in this MG. We observed load shedding when the MGC and MMGC were not able to provide the load.

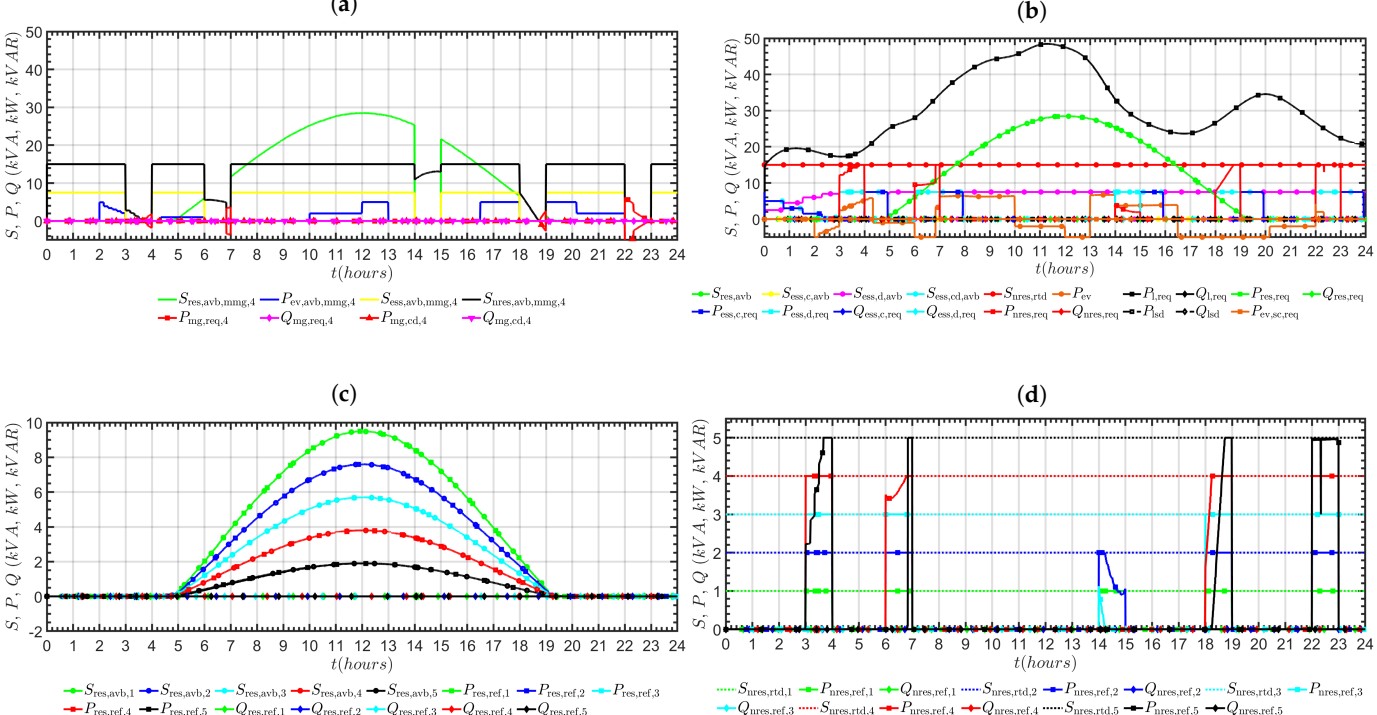

**Figure 6.** *Cont.*

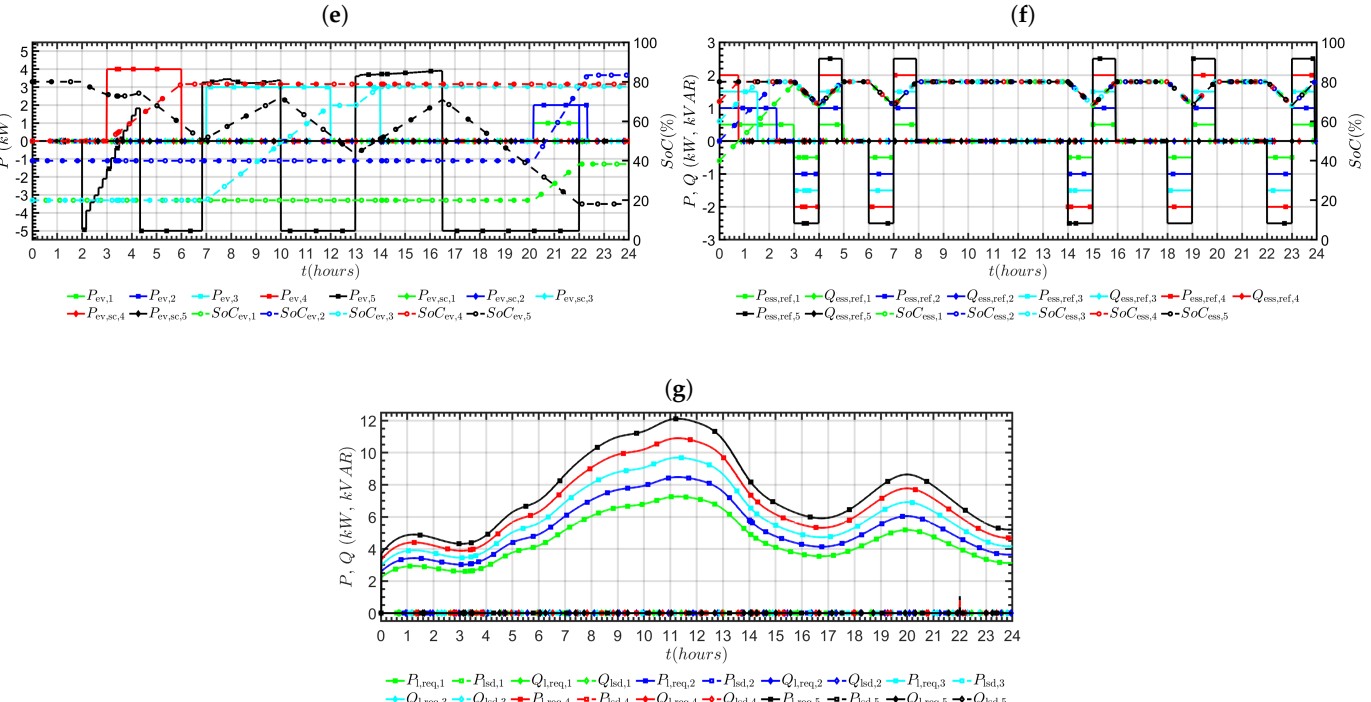

**Figure 6.** Behavior of MG 4 controllers. (**a**) MMGC for MG 4; (**b**) MGC 4; (**c**) RESC 4; (**d**) NRESC 4; (**e**) EVC 4; (**f**) ESSC 4; (**g**) LC 4.

### 3.5. Response of MG 5

Figure 7 represents the behavior of the different controllers for the fifth MG. This MG was identical to the first MG, with the exception that it had high RESs. Thus, all the loads were satisfied with no load shedding. Such a MG with excess resources took part in the MMG operation, as is evident from Figure 7a. The ESUs were discharged during islanding if required. Otherwise, the ESUs were kept fully charged. The NRESs remained idle in grid-tie mode and only operated in islanded conditions if required. The EVs are charged and discharged according to their demands with no power shedding or curtailment.

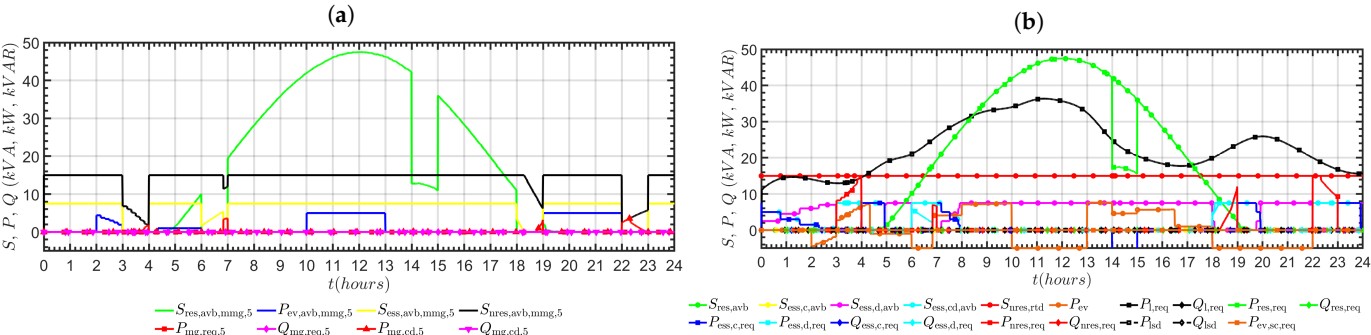

**Figure 7.** *Cont.*

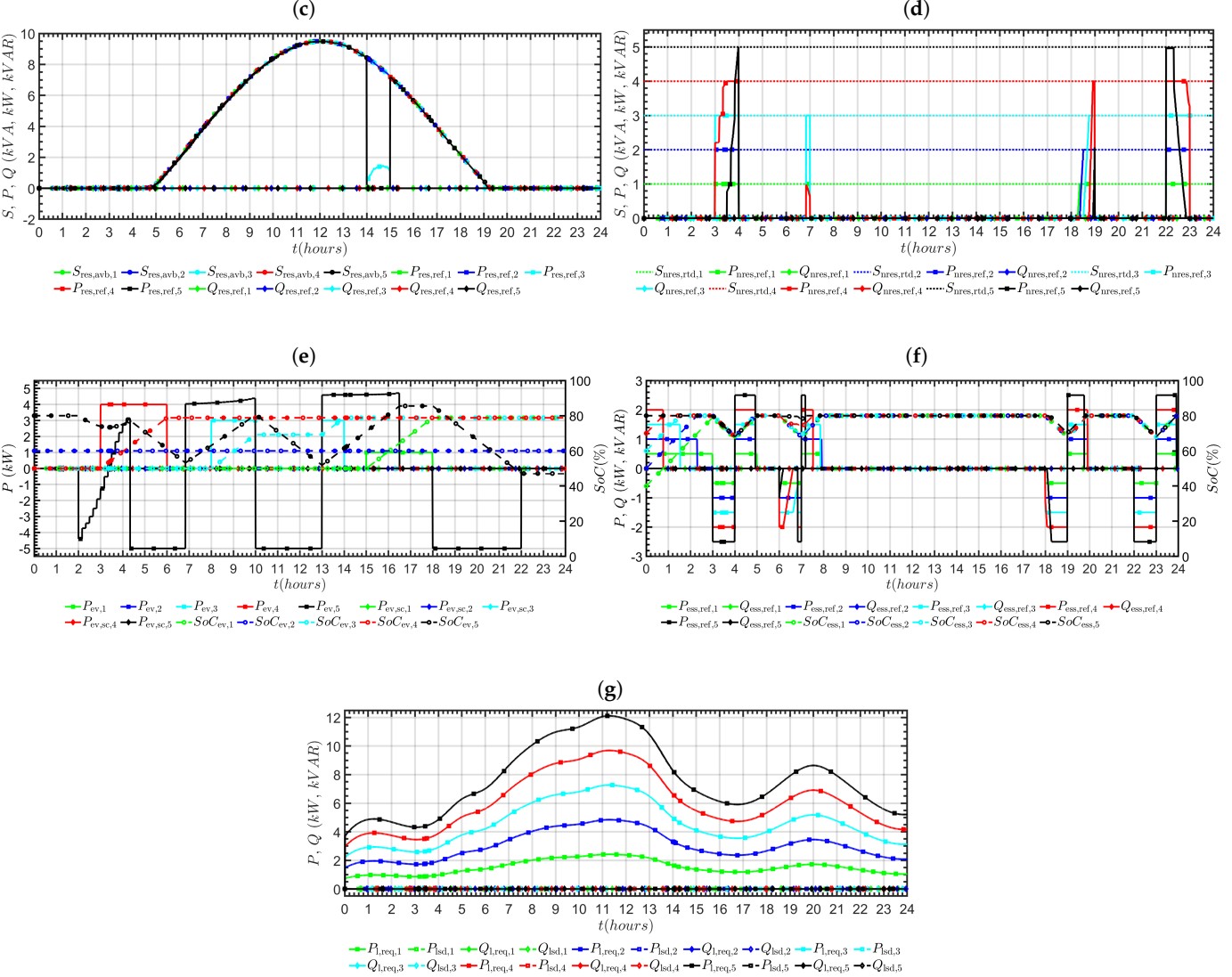

**Figure 7.** Behavior of MG 5 controllers. (**a**) MMGC for MG 5; (**b**) MGC 5; (**c**) RESC 5; (**d**) NRESC 5; (**e**) EVC 5; (**f**) ESSC 5; (**g**) LC 5.

## 4. Conclusions

A novel tetra-level dynamic decomposition based coordinated control of electric vehicles in multimicrogrids is proposed. The proposed control scheme is comprehensive, generic, modular, and secure in nature, with the intention to maximize renewable energy utilization, while meeting the required load. There are a number of microgrids that are connected to the grid. Each microgrid consists of renewable energy sources, energy storage systems, non-renewable energy sources, electric vehicles, and loads. Each distributed energy source or load is controlled by a microsource controller. All the microsource controllers with similar distributed energy sources are put under a unit controller. Thus, there are five unit controllers; namely, a renewable energy source unit controller, energy storage system unit controller, electric vehicle unit controller, non-renewable energy source unit controller, and load unit controller. The unit controllers in a microgrid are controlled by a microgrid controller. There is a single multimicrogrid controller at the top. Electric vehicles can operate in five modes of operation; namely, minimum time charging mode of operation, minimum time fixed cost charging mode of operation, minimum cost charging mode of operation, maximum state of charge fixed cost charging mode of operation,

and grid-support mode of operation. The proposed control scheme was verified using simulation-based case studies.

In future, this work could be extended to consider reactive power support from EVs. MMG control could be improved, to take care of MMG interactions during partial islanding; i.e., when some MGs are islanded and some are still grid connected.

**Author Contributions:** Data curation, M.A.A.; Formal analysis, M.A.A.; Funding acquisition, M.K.; Investigation, S.A.R.K.; Methodology, M.M.G.; Project administration, M.A.; Resources, S.A.R.K.; Supervision, S.A.R.K. and M.K.; Visualization, M.M.G.; Writing—review and editing, M.A. All authors have read and agreed to the published version of the manuscript.

**Funding:** The authors extend their appreciation to the Deanship of Scientific Research at King Khalid University for funding this work through large group Research Project under grant number RGP2/239/44. In addition, we would like to acknowledge the support received from Saudi Data and AI Authority (SDAIA) and King Fahd University of Petroleum and Minerals (KFUPM) under SDAIA-KFUPM Joint Research Center for Artificial Intelligence, Dhahran 31261, KSA.

**Institutional Review Board Statement:** Not applicable.

**Informed Consent Statement:** Not applicable.

**Data Availability Statement:** Not applicable.

**Conflicts of Interest:** The authors declare no conflict of interest.

## Abbreviations

The following abbreviations are used in this manuscript:

| | |
|---|---|
| BESS | Battery Based Energy Storage System |
| DES | Distributed Energy Source |
| ESS | Energy Storage System |
| ESSC | Energy Storage System Unit Controller |
| ESU | Energy Storage Unit |
| EV | Electric Vehicle |
| EVC | Electric Vehicle Unit Controller |
| LC | Load Unit Controller |
| MC | Microsource Controller |
| MG | Microgrid |
| MGC | Microgrid Controller |
| MMG | Multimicrogrid |
| MMGC | Multimicrogrid Controller |
| MPPT | Maximum Power Point Tracking |
| NRES | Non Renewable Energy Source |
| NRESC | Non Renewable Energy Source Unit Controller |
| PCC | Point of Common Coupling |
| RES | Renewable Energy Source |
| RESC | Renewable Energy Source Unit Controller |
| SoC | State of Charge |
| UC | Unit Controller |

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
