# Peer review of "A Novel Multi Level Dynamic Decomposition Based Coordinated Control of Electric Vehicles in Multimicrogrids"

_sustainability, doi:10.3390/su151612648_

Round 1

Reviewer 1 Report

Dear authors,
the work is quite complete, but I have a few questions to ask you:
- how were the optimisation problems (minimisation/maximisation) solved? What types of algorithms did you use?
- Were all the optimisation algorithms written as a minimisation/maximisation problem according to a variable 'x' (min_x or max_x), can you make this more explicit in the reports?
- Have you considered whether there is always a solution to the optimisation problems? What happens if there is none?
- how were the results obtained? are they simulations or experiments? what kind of environment did you use and what kind of software?
-the bibliography should be expanded on the state of the art of the whole context of energy management of electric systems. For example: https://doi.org/10.1016/j.isatra.2020.07.032 and https://doi.org/10.3390/en11113216

Minor changes:
- can you align all notations on the state of charge (SoC or SOC)?

Thank you

Reviewer 2 Report

A review of the article can be found in the attached file.

Reviewer 3 Report

The article deals with the control of an electric vehicle charging network powered by renewable energy sources. Now that the number of electric vehicles is steadily increasing, it is necessary to grapple with the efficiency of electric power grids. The classic ones based on fossil-fuel power plants, as well as those using water power, assume a one-way flow of energy: from the power plant to the consumer. If you connect a renewable energy source, a solar battery, a windmill, you need a lot of control dynamics in the control systems. And about them is this article. The authors proposed two control algorithms for such controllers. They presented the results in Figures 3-7, and they are interesting, but it should be weighed that these are simulations. Here, validation to the real object is missing. But this they can write in the next paper.  

Round 2

Reviewer 1 Report

Dear authors,

thank you for your answers